# SpotIt🔍: Evaluating Text-to-SQL Evaluation with Formal Verification

**Rocky Klopfenstein**[1][\*], **Yang He**[2][\*], **Andrew Tremante**[1][\*], **Yuepeng Wang**[2],
**Nina Narodytska**[3], **Haoze Wu**[1,3]
[1]Amherst College    [2]Simon Fraser University    [3]VMware Research by Broadcom
{rklopfenstein27,atremante26,hwu}@amherst.edu
{yha244,yuepeng}@sfu.ca, nina.narodytska@broadcom.com

## Abstract

Community-driven Text-to-SQL evaluation platforms play a pivotal role in tracking the state of the art of Text-to-SQL performance. The reliability of the evaluation process is critical for driving progress in the field. Current evaluation methods are largely test-based, which involves comparing the execution results of a generated SQL query and a human-labeled ground-truth on a static test database. Such an evaluation is optimistic, as two queries can coincidentally produce the same output on the test database while actually being different. In this work, we propose a new alternative evaluation pipeline, called SpotIt, where a formal bounded equivalence verification engine actively searches for a database that differentiates the generated and ground-truth SQL queries. We develop techniques to extend existing verifiers to support a richer SQL subset relevant to Text-to-SQL. A performance evaluation of ten Text-to-SQL methods on the high-profile BIRD dataset suggests that test-based methods can often overlook differences between the generated query and the ground-truth. Further analysis of the verification results reveals a more complex picture of the current Text-to-SQL evaluation.

## 1 Introduction

Text-to-SQL is one of the fundamental building blocks for designing natural language (NL) interfaces that enable users to access and analyze structured data sources. Translating human questions into executable database queries bridges the gap between non-technical users and complex data systems. This functionality underpins modern chatbots and smart assistants across a wide range of industrial applications, such as observability platforms for monitoring system health (BitsAI, 2025; Splunk, 2025), critical business processes (Amazon, 2025), and healthcare (Amazon Web Services, 2024).

Due to its practical relevance for commercial products, Text-to-SQL has recently attracted significant attention, leading to the development of a wide range of solutions (Shi et al., 2024). New Text-to-SQL frameworks are announced regularly, and thanks to community-driven evaluation platforms such as BIRD (Li et al., 2024) and Spider (Lei et al., 2024), their performance can be benchmarked and compared in near real time. Given the pivotal role these platforms play in tracking the state of the art, the reliability of their evaluation processes is crucial for driving progress in the field.

In this paper, we take a close look at the evaluation process for the accuracy of Text-to-SQL methods. Currently, the process usually involves checking whether the SQL queries generated by a method produce results equivalent to those of the *gold SQLs* (i.e., human-written ground-truth SQLs), under a pre-defined notion of equivalence. Most state-of-the-art evaluation frameworks (Li et al., 2024; Lei et al., 2024) perform this equivalence check through *testing*: executing both queries on a static test database and comparing the results. If the results match, the generated SQL is labeled as correct. Although widely used in practice, the testing-based approach has clear limitations. Because the check is performed on a single database, two different SQL queries may appear equivalent by chance, purely due to the specific data contained in that database. This raises an important question: when the test-based approach marks a generated SQL as correct, how often does it truly produce the same

---

[\*]Equal contribution.

results as the gold SQL in general? The next broader question is: to what extent can the current evaluation process accurately measure the performance of Text-to-SQL methods?

We investigate these questions by exploring an alternative correctness evaluation methodology. Instead of relying on test databases to assess equivalence, we propose to actively *search* for databases that can differentiate the generated SQL from the gold SQL. The search-based evaluation naturally provides stronger correctness guarantees and enables a more rigorous measurement of accuracy. Since providing complete equivalence guarantee is in general undecidable, we perform SMT-based bounded verification (He et al., 2024), which searches for differentiating databases with specified sizes. We develop a new Text-to-SQL evaluation workflow, SPOTIT, on top of those verification techniques. We significantly extend these techniques to support a new set of SQL operators over strings and dates which are commonly used for Text-to-SQL benchmarks.

Experiments on ten state-of-the-art Text-to-SQL methods on the popular BIRD dataset (Li et al., 2024) suggest that the reported accuracy of these methods drops by 11.3%–14.2% when switching from the official test-based evaluation to SPOTIT. The varying levels of decrease in absolute precision also lead to substantial changes in the order of ranking of the Text-to-SQL methods. Moreover, SPOTIT produces minimal differentiating databases, which enables us to pinpoint the sources of inconsistencies between the generated and gold SQLs. Analysis of these databases uncovers several shortcomings of the current Text-to-SQL evaluation process. Most surprisingly, we find that when the predicted SQL disagrees with the ground truth, it is often the gold SQL that is incorrect.

To summarize, our contributions include:

- SPOTIT, a new evaluation pipeline for Text-to-SQL powered by formal equivalence verification;
- novel SMT-encoding for a set of SQL operators over strings and dates, and proof of its correctness;
- practical strategies for the efficient deployment of SPOTIT;
- a large-scale evaluation of ten state-of-the-art Text-to-SQL methods on the BIRD dataset, which reveals several potential shortcomings of current Text-to-SQL evaluation.

## 2 PRELIMINARIES

We provide background on Text-to-SQL and formal equivalence checking. Due to space limitation, an overview of related work is present in App. A.

**Text-to-SQL problem statement.** Given a natural language query $N$ and a database $D$ with schema $\mathcal{S}$, the goal of Text-to-SQL is to map $(N, D)$ to an SQL query $Q$, such that executing $Q$ on $D$, denoted $Q(D)$, produces an output relation (table) that answers $N$.

**Text-to-SQL evaluation.** The main evaluation mechanism for a Text-to-SQL framework relies on a gold SQL query produced by a human annotator. Hence, for each natural language query $N$ over a database, there exists a gold SQL query $Q$ that represents the human-labelled ground truth of translating $N$ into SQL. Given a generated SQL $P$ and the corresponding gold query $Q$, current evaluation performs the following check:

$$\text{EX-TEST}(P, Q, D_{\text{test}}) = \begin{cases} 1, & \text{if } \forall r.\ r \in P(D_{\text{test}}) \leftrightarrow r \in Q(D_{\text{test}}) \\ 0, & \text{otherwise,} \end{cases} \tag{1}$$

where $D_{\text{test}}$ is a test database provided by the benchmark set, and $r$ denotes a row in the result table. In words, EX-TEST compares whether the two tables, $P(D_{\text{test}})$ and $Q(D_{\text{test}})$, contain the same set of rows. In order to more rigorously analyze the equivalence between $P$ and $Q$, we use formal verification to search for a differentiating database $D_{\text{cex}}$ such that EX-TEST$(P, Q, D_{\text{cex}}) = 0$.

**Bounded SQL equivalence checking.** Given two SQL queries $Q_1$ and $Q_2$ over a schema $\mathcal{S}$ and an upper bound $K$ on the relation size, the problem of bounded equivalence checking is to decide whether $Q_1$ and $Q_2$ are equivalent, denoted $Q_1 \simeq_{\mathcal{S},K} Q_2$, for all databases $D$ conforming to $\mathcal{S}$ such that each relation in $D$ has at most $K$ tuples. Formally,

$$Q_1 \simeq_{\mathcal{S},K} Q_2 \stackrel{\text{def}}{=} \forall D \in \text{Instances}(\mathcal{S}).\ \forall R \in \text{Relations}(D).\ |R| \leq K \Rightarrow Q_1(D) = Q_2(D),$$

where Instances($\mathcal{S}$) represents all database instances conforming to $\mathcal{S}$, and Relations($D$) represents all relations in $D$. In general, the goal is either to prove the bounded equivalence holds, or to find a counterexample database $D_{\text{cex}}$ that disproves the equivalence. Compared with unbounded equivalence

$N_1$: *"Which is the youngest patient with an abnormal anti-ribonuclear protein level?*
*Please list his or her date of birth."*

```
/*Gold SQL Q*/:
SELECT T1.birthday
FROM patient AS T1
INNER JOIN laboratory AS T2
ON T1.ID = T2.ID
WHERE T2.rnp != '-' OR '+-'
ORDER BY T1.birthday DESC LIMIT 1
/*Generated SQL P*/:
SELECT patient.birthday
FROM patient
INNER JOIN laboratory
ON patient.ID = laboratory.ID
WHERE NOT laboratory.rnp IN ('-', '+-')
ORDER BY patient.birthday
DESC LIMIT 1
```

$N_2$: *"How many male patients who underwent testing between 1995 and 1997 and were subsequently diagnosed with Behcet disease did not stay in the hospital for treatment?"*

```
/*Gold SQL Q*/:
SELECT COUNT(T1.id) FROM patient AS T1
INNER JOIN examination AS T2 ON T1.id = T2.id
WHERE T2.diagnosis = 'Behcet'  AND T1.sex = 'M'
AND STRFTIME('%Y', T2.examination_date)
BETWEEN '1995' AND '1997' AND T1.admission = '-';
/*Generated SQL P*/:
SELECT COUNT(DISTINCT patient.id)
FROM patient INNER JOIN examination
ON patient.id = examination.id
WHERE patient.sex = 'M' AND
examination.examination_date
BETWEEN '1995-01-01' AND '1997-12-31'
AND examination.diagnosis = 'Behcet'
AND patient.admission = '-';
```

Figure 1: Examples of cases where the generated SQL produces the same output as the gold SQL on the BIRD's official test database, but SPOTIT finds a database that differentiates the the queries. The parts that explain the mismatch are highlighted. For $N_1$, the gold SQL is incorrect. And for $N_2$, both SQL queries can be right depending on the interpretation of the NL question.

checking, which is generally undecidable (Mohamed et al., 2024), bounded equivalence checking can handle a more expressive SQL subset and is guaranteed to uncover small counterexamples (if they exist). These features make bounded verification suitable for large-scale Text-to-SQL evaluation.

**VERIEQL.** VERIEQL (He et al., 2024) is a recently proposed bounded equivalence checker for SQL queries and, to the best of our knowledge, supports the most expressive subset of SQL among existing tools. It reduces the verification task to a satisfiability problem by encoding the symbolic execution of the two SQL queries and the *non-equivalence* of the execution results as a satisfiability modulo theories (SMT) formula (Barrett & Tinelli, 2018), which can be solved by an off-the-shelf SMT solver (De Moura & Bjørner, 2008). The bounded equivalence property holds if and only if the formula is unsatisfiable, which means it is not possible to find a database that result in different execution results. Otherwise, a satisfying interpretation of the formula can be decoded to a counterexample database. We significantly extend VERIEQL to support our verification use cases.

## 3 MOTIVATING EXAMPLES

Before we describe our new verification-based evaluation pipeline, we first discuss main sources of mismatches between the gold SQL and the generated SQL in Text-to-SQL evaluation. There are three main such sources: (1) NL query $N$ is ambiguous, so both the gold and generated SQL queries are justifiable interpretations; (2) $N$ is unambiguous, but the gold SQL query is incorrect (gold SQLs are created manually and thus prone to human errors); (3) $N$ is unambiguous, the gold SQL query is correct, but the generated SQL query is incorrect. Our framework focuses on checking equivalence between the gold SQL and the generated SQL, treating the latter as the best-effort, semantically correct formalization of $N$. We show that SPOTIT can successfully detect incorrect generated SQLs that are overlooked by existing test-based evaluation. Perhaps more surprisingly, SPOTIT also allows us to spot the first and second sources of mismatch. Fig. 1 shows two illustrative examples.

**Example 3.1.** *Consider the query* $N_1$: *"Which is the youngest patient with an abnormal anti-ribonuclear protein level? Please list his or her date of birth." together with the gold and generated SQL queries. On the development database that BIRD provides, both queries return "1989-08-28". However,* SPOTIT *found a database on which these two queries are not equivalent (Appendix D.1). In fact, we observe that all ten frameworks that we tested generated SQLs that are not equivalent to the gold query. Upon closer inspection, we find that the gold query is incorrect: its* `WHERE` *clause is equivalent to* `T2.rnp != '-' OR FALSE`, *as a string literal like* `'+-'` *is interpreted as* `FALSE` *in a boolean context, which is not the intended behavior.* □

**Example 3.2.** *Consider another query* $N_2$: *"How many male patients who underwent testing between 1995 and 1997 and were subsequently diagnosed with Behcet disease did not stay in the hospital for treatment?" together with the gold and generated SQL queries. These two queries both return "2" on the BIRD test database. However, the two queries are clearly not equivalent (*`id` *is not a primary key of the* `examination` *table therefore duplicates are allowed): the generated query counts all*

$$
\begin{array}{rcl}
\text{Query } Q_r & ::= & Q \mid \text{OrderBy}(Q, \vec{E}, b) \\
\text{Subquery } Q & ::= & R \mid \Pi_L(Q) \mid \sigma_\phi(Q) \mid \rho_R(Q) \mid Q \oplus Q \mid \text{Distinct}(Q) \mid Q \otimes Q \mid \text{GroupBy}(Q, \vec{E}, L, \phi) \mid \text{With}(\vec{Q}, \vec{R}, Q) \\
\text{Attr List } L & ::= & id(A) \mid \rho_a(A) \mid L, L \\
\text{Attr } A & ::= & \text{Cast}(\phi) \mid E \mid \mathcal{G}(E) \mid A \diamond A \\
\text{Pred } \phi & ::= & b \mid \text{Null} \mid A \odot A \mid \text{IsNull}(E) \mid \vec{E} \in \vec{v} \mid \vec{E} \in Q \mid \phi \wedge \phi \mid \phi \vee \phi \mid \neg\phi \\
& \mid & \textbf{PrefixOf}(s, E) \mid \textbf{SuffixOf}(s, E) \mid \textbf{Like}(s, E) \mid \textbf{Contain}(s, E) \\
\text{Expr } E & ::= & a \mid v \mid E \diamond E \mid \text{ITE}(\phi, E, E) \mid \text{Case}(\vec{\phi}, \vec{E}, E) \mid \textbf{SubStr}(E_1, E_2, E_3) \mid \textbf{Concat}(E_1, E_2) \\
& \mid & \textbf{Strftime}(\kappa, E) \mid \textbf{JulianDay}(E) \mid \textbf{DateShift}(E, i, \delta) \mid \textbf{ToInt}(E) \mid \textbf{ToDate}(E) \mid \textbf{ToStr}(E) \\
\text{Join Op } \otimes & ::= & \times \mid \bowtie_\phi \mid {\rtimes}_\phi \mid {\ltimes}_\phi \mid {\bowtie\!\!\!\!\times}_\phi \\
\text{Collection Op } \oplus & ::= & \cup \mid \cap \mid \setminus \mid \uplus \mid \Cap \mid - \\
\text{Arith Op } \diamond & ::= & + \mid - \mid \times \mid / \mid \% \\
\text{Logic Op } \odot & ::= & \leq \mid < \mid = \mid \neq \mid > \mid \geq
\end{array}
$$

$R \in$ Relation Names $\quad a \in$ Attribute Names $\quad v \in \{\text{Null}\} \cup \textbf{Integers} \cup \textbf{Dates} \cup \textbf{Strings} \quad b \in$ Bools $\quad i \in \textbf{Integers}$
$s \in \textbf{Strings} \quad \mathcal{G} \in \{\text{Count, Min, Max, Sum, Avg}\} \quad \kappa \in \{\textbf{"%Y", "%M", "%d"}\} \quad \delta \in \{\textbf{"Year", "Month", "Day"}\}$

Figure 2: Extended syntax of SQL Queries. New features are in bold.

*examinations per patient, whereas the gold query counts only distinct patients. SPOTIT easily found a database that differentiate the two queries (Appendix D.2). Note that depending on the interpretation of the question, both SQL queries can be correct: the gold SQL can be reasonable if the goal is to understand the hosptial workload, while the generated SQL can be reasonable if the goal is to understand the number of unique patients. Hence, we conclude that $N_2$ is ambiguous.* $\qquad\square$

Note that these examples were overlooked by existing test-based evaluations. On the other hand, using SPOTIT, we found that undetected cases like those are quite common in the BIRD dataset.

## 4  METHODOLOGY

In this section, we introduce new SMT-encodings for a number of SQL operators over string and date types that were not supported by existing bounded equivalence verification methods but frequently appear in Text-to-SQL benchmarks. Then we present our verification-based evaluation pipeline SPOTIT and discuss practical implementation strategies.

### 4.1  EQUIVALENCE CHECKING FOR SQL QUERIES

To understand our extension, let us first walk through Example 4.1 to understand how equivalence checking can be encoded as an SMT formula in a verifier like VERIEQL (He et al., 2024).

**Example 4.1.** *Consider a schema $\mathcal{S} = \{R \mapsto \{id:\text{int}, dob:\text{date}\}\}$ and the following two queries:*

$Q_1 = $ SELECT *id* FROM *R* WHERE *id>1* $\qquad\qquad Q_2 = $ SELECT *id* FROM *R* WHERE *id>2*

*We describe how to encode equivalence checking for a bound $(K)$ of 1 as an SMT formula. First, variables are introduced to represent the database and the execution results. This includes a symbolic database $D = \{R \mapsto [t_1]\}$, where $t_1 = [x_1, x_2]$ is a tuple in $R$, and $x_1, x_2$ are integer variables. In addition, tuples $t_2 = [x_3]$ and $t_3 = [x_4]$, are introduced to encode query results: $Q_1(D) = [t_2]$ and $Q_2(D) = [t_3]$, where $x_3, x_4$ are both integer variables. Note that the number of tuples in $R$ is equal to the bound $K$. Also note that a date $(x_2)$ is represented as an integer, which is sufficient here but not in general. We later introduce precise encoding of date to support richer operations.*

*We now describe the constraints over the variables. The first set of constraints ensures that $t_2$ and $t_3$ correctly capture the semantics of $Q_1$ and $Q_2$. In this case, $t_2$ tuple is constrained by $\Phi_{Q_1} = (x_1 > 1 \rightarrow (x_3 = x_1 \wedge \neg Del(t_2))) \wedge (x_1 \leq 1 \rightarrow Del(t_2))$, where Del is an uninterpreted function denoting the non-existence of a symbolic tuple. The formula $\Phi_{Q_1}$ ensures that only interpretations satisfying $x > 1$ can populate a concrete tuple; otherwise, $Q_1$'s result is empty. Similarly, $t_3$ is constrained by $\Phi_{Q_2} = (x_1 > 2 \rightarrow (x_4 = x_1 \wedge \neg Del(t_3))) \wedge (x_1 \leq 2 \rightarrow Del(t_3))$.*

*The second set of constraints encodes that $Q_1(D)$ and $Q_2(D)$ returns different results. In this case, it is simply $t_2 \neq t_3$. The full encoding is a conjunction of all constraints: $\Phi_{Q_1} \wedge \Phi_{Q_2} \wedge (t_2 \neq t_3)$, whose satisfiability can be checked by an SMT solver. A satisfying interpretation to this conjunction corresponds to a database instance that differentiates $Q_1$ and $Q_2$. For example, the queries are not equivalent under the interpretation $\mathcal{I} = \{x_1 \mapsto 2\}$.* $\qquad\square$

**Extension in SQL encoding.** Existing bounded SQL equivalence checker still lacks support for several important features, including precise encoding of dates and strings, which are highly relevant

in Text-to-SQL applications. Furthermore, SQL supports computations across many different data types with implicit type casting (e.g., 1 + "a" and date("2000-01-01") + "1"), which poses significant challenges to establish precise semantics and encodings. To address these limitations and challenges, we introduce techniques to support dates and strings, along with their manipulations, in the SQL equivalence checker VERIEQL. We also introduce type conversions across Null, integers, dates, and strings for implicit type casting. For example, in the gold SQL for $N_2$ (Fig. 1), the output of the STRFTIME function is implicitly converted from a date to an integer.

Fig. 2 presents our supported SQL grammar. Specifically, the query language introduces type conversions among various data types (e.g., ToInt($E$), ToDate($E$), and ToString($E$)), which allows us to precisely establish the semantics of dates and strings and enhances the expressiveness of our SQL subset. We also incorporate additional expressions and predicates for data and string manipulations, such as date formatting Strftime($\kappa, E$), Julian day JulianDay($E$), string pattern matching PrefixOf($s, E$), SuffixOf($s, E$), Like($s, E$), and string truncation SubStr($E_1, E_2, E_3$). The symbolic encoding for these extended expressions and predicates is formally presented in Appendix F.

As an example, we describe how to precisely encode a date variable, which is very common in Text-to-SQL. For instance, the date of birth and the time of a transaction are naturally modeled with the date type. Previously, date was encoded as a single integer variable (see Example 4.1). Although this coarse representation still enables the encoding of certain date operations (e.g., comparison), it does not necessarily support all date operations, such as date-formatting, which is used in the gold SQL query for $N_2$ in Fig. 1. As a date can be viewed as a triplet (year, month, day), we introduce three integer variables $y$, $m$, and $d$, and constrain their values with the following formula $\Phi$:

$$\Phi = \Phi_1 \wedge \Phi_2 \wedge \Phi_3, \text{ where } \Phi_1 = \text{MIN\_YEAR} \leq y \leq \text{MAX\_YEAR}, \Phi_2 = 1 \leq m \leq 12,$$
$$\Phi_3 = 1 \leq d \wedge (\vee_{c \in \{1,3,5,7,8,10,12\}} m = c \rightarrow d \leq 31)$$
$$\wedge (m = 2 \rightarrow d \leq 28 + \text{ite}(leap(y), 1, 0)) \wedge (\vee_{c \in \{4,6,9,11\}} m = c \rightarrow d \leq 30)$$

The term $leap(y)$ encodes the leap year condition: $y \% 4 = 0 \wedge (y \% 100 \neq 0 \vee y \% 400 = 0)$. Constraints $\Phi_1$, $\Phi_2$, and $\Phi_3$ restrict the possible values of the year, the month, and the day, respectively. For example $\Phi_1$ specifies the valid range of the year, which is specific to the database engine. For example, SQLite only accepts dates between "0000-01-01" and "9999-12-31"; in which case MIN\_YEAR is 0 and MAX\_YEAR is 9999. This refined representation allows us to precisely encode a rich set of date operations and analyze more SQL queries compared to the previous encoding.

**Equivalence under set semantics.** SQL equivalence checkers typically support equivalence under bag semantics and list semantics. However, some Text-to-SQL evaluation platforms, such as BIRD (Li et al., 2024), by default adopt equivalence under set semantics (see equation 1). This can be expressed as an SMT constraint. Given two query results with symbolic tables $R_1 = [t_1, \ldots, t_n]$ and $R_2 = [r_1, \ldots, r_m]$, the condition that $R_1$ and $R_2$ are equivalent under set semantics is as follows:

$$\bigwedge_{i=1}^{n} \left(\neg\text{Del}(t_i) \rightarrow \vee_{j=1}^{m}(\neg\text{Del}(r_j) \wedge t_i = r_j)\right) \wedge \bigwedge_{j=1}^{m} \left(\neg\text{Del}(r_j) \rightarrow \vee_{i=1}^{n}(\neg\text{Del}(t_i) \wedge r_j = t_i)\right) \quad (2)$$

On a high level, equivalence is defined by mutual set containment: $R_1 = R_2$ iff $R_1 \subseteq R_2$ and $R_2 \subseteq R_1$. But since some tuples might be deleted due to WHERE clauses, we restrict set containment to non-deleted tuples, i.e., those satisfying $\neg\text{Del}(t)$.

**Correctness of the encodings.** We now state the correctness of our symbolic encoding for the extended expressions and predicates, as well as the equivalence under set semantics. Proof of these theorems is in Appendix G. As we encode the symbolic execution of queries, to prove the correctness of our approach, we need to show that our symbolic execution coincides with the concrete execution. This involves showing that given an expression $E$, the satisfying interpretation of $E$'s symbolic execution result is identical to the concrete execution result of $E$. Thm. 1 states that formally.

**Theorem 1** (Correctness of expression encoding). *Let $D$ be a database over schema $\mathcal{S}$, $xs$ be a tuple list, and $E$ be an expression. Consider a symbolic database $\Gamma$ over $\mathcal{S}$, a list of symbolic tuples $\mathcal{T}$, and $E$'s symbolic encoding $[\![E]\!]_{\mathcal{S},\Gamma,\mathcal{T}}$. For any satisfying interpretation $\mathcal{I}$ with $\mathcal{I}(\Gamma) = D \wedge \mathcal{I}(\mathcal{T}) = xs$, evaluating the expression $E$ over the database $D$ and the tuple list $xs$ yields the interpretation of $E$'s symbolic encoding $\mathcal{I}([\![E]\!]_{\mathcal{S},\Gamma,\mathcal{T}})$, i.e., $\mathcal{I}(\Gamma) = D \wedge \mathcal{I}(\mathcal{T}) = xs \Rightarrow [\![E]\!]_{D,xs} = \mathcal{I}([\![E]\!]_{\mathcal{S},\Gamma,\mathcal{T}})$.*

Similarly, given a predicate $\phi$, the satisfying interpretation of $\phi$'s symbolic execution result is also identical the concrete execution result of $\phi$. This is formally stated in Appendix G. Lastly, we state the correctness of our encoding for equivalence under set semantics.

**Algorithm 1** Bounded equivalence checking

**Require:** Database schema $\mathcal{S}$, gold SQL query $Q$, generated SQL query $P$, time limit $T$, bound $K$
**Ensure:** A counterexample $D_{\text{cex}}$
1: **function** EQUIVCHECK($\mathcal{S}, Q, P, T, K$)
2:     **for** $k \in [1, K]$ **do**
3:         res, $D_{\text{cex}} \leftarrow$ CHECKBOUND($\mathcal{S}, P, Q, k, T$)
4:         **if** res $=$ EQUIVALENT **then continue**
5:             ▷ Bounded equivalence under $k$
6:         **else if** res $=$ NON-EQUIVALENT **then**
7:             ▷ Find a counterexample
8:             ▷ Validate the counterexample on the backend DBMS
9:             **if** ¬EX-TEST($P, Q, D_{\text{cex}}$) **then**
10:                 **return** $\{D_{\text{cex}}\}$
11:         **else break** ▷ Timeout, unsupported, undecidable queries
12:     **return** $\emptyset$

**Algorithm 2** SPOTIT⁺

**Require:** Database $\mathcal{S}$, user query $N$, gold SQL query $Q$, Text-to-SQL frameworks $\mathcal{M}$, time limit $T$ and bound $K$
**Ensure:** Counterexamples $D_{\text{cexs}}$
1: **function** SPOTIT⁺($\mathcal{S}, N, \mathcal{M}, T, K$)
2:     $D_{\text{cexs}} \leftarrow \emptyset$
3:     **for** $m \in \mathcal{M}$ **do**
4:         $P \leftarrow m(\mathcal{S}, N)$     ▷ Generate SQL query $P$ using $m$
5:         $D_{\text{cexs}}[m] \leftarrow$ EQUIVCHECK($\mathcal{S}, Q, P, T, K$)
6:     ▷ Performing cross-referencing counterexamples
7:     $D^*_{\text{cexs}} \leftarrow \cup_{m \in \mathcal{M}} D_{\text{cexs}}[m]$
8:     **for** $m \in \mathcal{M}$ **do**
9:         **for** $D \in D^*_{\text{cexs}} \setminus D_{\text{cexs}}[m]$ **do**
10:             **if** ¬EX-TEST($P, Q, D$) **then**
11:                 $D_{\text{cexs}}[m] \leftarrow D_{\text{cexs}}[m] \cup \{D\}$
12:     **return** $D_{\text{cexs}}$

**Theorem 2** (Equivalence under set semantics). *Given two relations $R_1 = [t_1, \ldots, t_n]$ and $R_2 = [r_1, \ldots, r_m]$, if formula (2) is valid, then $R_1$ and $R_2$ are equivalent under set semantics.*

## 4.2 SPOTIT: A SEARCH-BASED TEXT-TO-SQL EVALUATION PIPELINE

Fig. 3 presents a high-level workflow of our approach that consists of three conceptual phases.

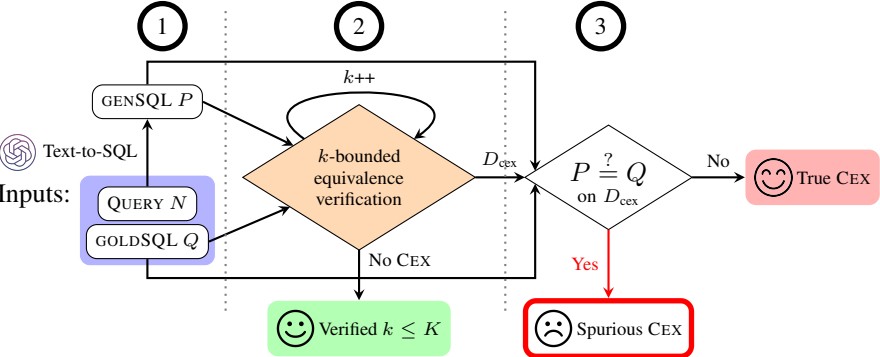

Figure 3: Three main phases of SPOTIT.

① **Input phase.** Given a NL question $N$ and its corresponding gold SQL query $Q$, a Text-to-SQL framework takes as input $N$ and generates a SQL query $P$. Both $Q$ and $P$ are passed to phase ②.

② **Verification phase.** The goal is to find a counterexample database instance on which the queries $Q$ and $P$ produce different outputs. For a given bound $k \leq K$, we perform bounded equivalence checking between $Q$ and $P$. If the queries are proved equivalent, then we increase $k$ by one for the next verification check. Furthermore, we cannot find any counterexample under all bounds and conclude that they are verified up to the bound $k$. On the other hand, if the queries are proved to be non-equivalent under some bound, we proceed to phase ③ for a further validation of $D_{\text{cex}}$.

③ **Validation phase.** Given the queries $Q$ and $P$ and a counterexample $D_{\text{cex}}$ returned by verification algorithm, we must verify that this counterexample is non-spurious. There are two main reasons spurious counterexamples can arise in the verification engine. Either because some operators are over-approximated in the SMT encoding or the SQL query admits non-deterministic behaviors that cannot be modeled. Therefore, we execute the queries on the counterexample database (e.g., in SQLite) and check whether the results actually differ. $D_{\text{cex}}$ is viewed valid if the results remain different; otherwise, we report this spurious case to the developers.

Alg.1 implements the second and third phases. For a given bound $k \leq K$, it first checks bounded equivalence between $Q$ and $P$ (line 3). If the queries are proven to be non-equivalent (line 6) under some bound, we validate that the counterexample database is indeed a true counterexample (line 9) and return it if this is the case. If the queries are proven to be equivalent in line 3, then we increase $k$ by one for the next verification step. If the verifier cannot find any counterexample under all bounds,

Alg.1 returns an empty set. Finally, if the verifier times out on a bound $k$, or the query is unsupported or undecidable, it also returns an empty set.

**Cross-checking counter-examples.** One observation we make is that as we progress through the frameworks, we collect a set of counterexamples that separate the gold query from the generated queries. Hence, we realized that these counterexamples can be reused as checks across all frameworks, as they might generalize across frameworks. Alg.2 implements this idea. First, it obtains counterexample databases, if they exist, for all frameworks by calling Alg.1 (lines 3–5). Then, it iterates over all frameworks again and tests equivalence between $Q$ and $P$ on these counterexample databases (lines 7–11). Empirically, this improves the effectiveness of our approach.

## 5 EXPERIMENTAL EVALUATION

In this section, we investigate the effect of using SPOTIT as the evaluation methodology for Text-to-SQL tasks. We are interested in the following questions:

- How much more SQL queries does our extension of VERIEQL support?
- Can SPOTIT provide more rigorous accuracy evaluation than test-based approaches?
- Can SPOTIT reveal shortcomings in existing Text-to-SQL evaluations?

**Experimental Setup.** We consider all 1,533 question-SQL pairs from the development set of BIRD (Li et al., 2023b), a state-of-the-art dataset for evaluating Text-to-SQL methods. The questions span 11 different databases from different professional domains, such as education, healthcare, and sports. The official BIRD leaderboard [1] contains over 80 Text-to-SQL methods and are updated frequently. Not all methods are open-source or have predictions publicly available. Therefore, we reached out to the developers of top-performing Text-to-SQL frameworks on the BIRD leaderboard and obtained the generated SQL queries for 10 of them, which constitutes a representative subset of state-of-the-art Text-to-SQL methods. The methods are listed in Tab. 1.

We first evaluate the predictions of each method using BIRD's official test-based execution accuracy metric (EX-TEST), which, as described in Eq. 1, compares the results of executing the generated and gold queries on a given test database. For predictions that are deemed correct by EX-TEST, we apply SPOTIT to perform a more rigorous analysis. We implemented SPOTIT on top of VERIEQL (He et al., 2024), which we extended using the methods described in Sec. 4.1. To generate practically relevant counterexamples, we also extend the verification condition to exclude degenerate counterexamples that result in empty for one SQL and NULL for the other SQL.

Table 1: The Text-to-SQL methods we evaluated

| Entry | Acronym |
|---|---|
| Alpha-SQL + Qwen2.5-Coder-32B (Li et al., 2025a) | ALPHA |
| CSC-SQL + Qwen2.5-Coder-7B (Sheng & Xu, 2025a) | CSC-7B |
| CSC-SQL + XiYanSQL (Sheng & Xu, 2025a) | CSC-32B |
| GenaSQL-1 (Dönder et al., 2025) | GENA-1 |
| GenaSQL-2 (Dönder et al., 2025) | GENA-2 |
| RSL-SQL + GPT-4o (Cao et al., 2024) | RSL |
| OmniSQL-32B (Li et al., 2025b) | OMNI-MAJ |
| GSR (anonymous authors) | GSR |
| CHESS$_{IR+CG+UT}$ (Talaei et al., 2024a) | CHESS |
| SLM-SQL + Qwen2.5-Coder-1.5B Sheng & Xu (2025c) | SLM |

We consider three variants of SPOTIT: (i) SPOTIT: Alg. 2 instantiated with the extended verification engine but without cross-checking (lines 7–11); (ii) SPOTIT⁻: Alg. 2 instantiated with vanilla VERIEQL and without cross-checking; (iii) SPOTIT⁺: Alg. 2 with cross-checking. We verify each generated-gold SQL pair up to a bound ($K$) of 5. Each verifier call is given one physical core, 8GB memory, and a CPU timeout of 600 seconds. In practice, a counterexample can typically be found within seconds, as reported below. Experiments were performed on a cluster equipped with Dell PowerEdge R6525 CPU servers featuring 2.6-GHz AMD CPU cores.

**Performance of Verification Engine.** We first evaluate the effect of our extensions to the original VERIEQL engine (He et al., 2024) in terms of *coverage*, defined as the fraction of generated-gold-SQL pairs that can be encoded into an SMT query. In addition, we measure the average runtime of SPOTIT on questions where a valid differentiating database is found. The results are shown in Tab. 2. Our extensions significantly increase the coverage of the verification engine on relevant questions (i.e., ones deemed correct by EX-TEST) for each method, allowing us to formally analyze a larger number of generated SQL queries. For example, for CSC-32B, the coverage increases from 84.83% to 94.88%, which corresponds to 110 additional supported questions $((94.88\% - 84.83\%) * 1094)$.

---

[1] https://bird-bench.github.io/

Table 2: % of SQL pairs supported by SPOTIT⁻ and SPOTIT. For SPOTIT, also the average time in seconds on pairs where CEXs are found by the verifier and how many of them are non-spurious.

| Method (# quest.) | SPOTIT⁻ (%) | SPOTIT (%) | Avg. Time | Valid. (%) |
|---|---|---|---|---|
| ALPHA (1064) | 84.87 | 93.89 | 3.10 | 96.15 |
| CHESS (976) | 87.40 | 97.13 | 1.40 | 93.34 |
| CSC-32B (1094) | 84.83 | 94.88 | 3.24 | 94.46 |
| CSC-7B (1061) | 85.77 | 96.14 | 3.93 | 95.10 |
| GENA-1 (1062) | 84.56 | 94.92 | 1.01 | 94.52 |
| GENA-2 (1082) | 84.47 | 94.55 | 0.93 | 95.42 |
| GSR (1020) | 84.51 | 93.63 | 1.12 | 94.86 |
| OMNI-MAJ (1026) | 86.65 | 95.61 | 1.36 | 95.83 |
| RSL (1038) | 86.03 | 95.18 | 1.64 | 95.62 |
| SLM (973) | 85.92 | 94.24 | 1.36 | 95.05 |

Table 3: Performance of Text-to-SQL methods using EX-TEST, EX-SPOTIT, and EX-SPOTIT⁺ on the 1533 BIRD-dev benchmarks.

| | EX-TEST | | EX-SPOTIT | | EX-SPOTIT⁺ | |
|---|---|---|---|---|---|---|
| | Acc. (%) | Rank | Acc.(%) | Rank | Acc.(%) | Rank |
| CSC-32B | 71.32 | 1 | 58.80 | 3 | 57.82 | 4 |
| GENA-2 | 70.53 | 2 | 59.84 | 1 | 59.13 | 1 |
| ALPHA | 69.36 | 3 | 55.87 | 6 | 55.02 | 6 |
| GENA-1 | 69.23 | 4 | 59.45 | 2 | 59.00 | 2 |
| CSC-7B | 69.17 | 5 | 58.54 | 4 | 57.95 | 3 |
| RSL | 67.67 | 6 | 56.58 | 5 | 55.80 | 5 |
| OMNI-MAJ | 66.88 | 7 | 54.69 | 7 | 54.04 | 7 |
| GSR | 66.49 | 8 | 54.56 | 8 | 53.72 | 8 |
| CHESS | 63.62 | 9 | 52.87 | 9 | 52.35 | 9 |
| SLM | 63.43 | 10 | 51.37 | 10 | 50.98 | 10 |

The average time taken by SPOTIT to find a counterexample is under 4 seconds for all methods, which, combined with the fact that the analysis for each question can be done in parallel, confirms that SPOTIT is already a practical method for formally comparing generated SQLs with gold SQLs. Moreover, a high percentage (up to 96.15%) of the counterexamples found by the verifier are successfully validated, which suggests that our SMT encoding is sufficiently precise in practice.

**Comparing test-based evaluation with SPOTIT.** We now evaluate the accuracy of each Text-to-SQL method based on EX-TEST, EX-SPOTIT, and EX-SPOTIT⁺. As shown in Tab. 3, the accuracy of each method drops significantly when SPOTIT is used to check query equivalence. For example, the accuracy of CSC-32B drops from 71.32% to 58.80% with SPOTIT, and further to 57.82% when cross-checking is enabled. This means that there are 207 generated SQLs ($1533 * (71.32\% - 57.82\%)$) that passed the test on the official test databases, but were differentiated from the gold SQL by SPOTIT. Overall, SPOTIT resulted in a decrease in accuracy ranging from 9.8% to 13.5%, and cross-checking results in a small further decrease, by up to 1%. Interestingly, the ranking of the Text-to-SQL methods also changes substantially when evaluated under the more stringent, verification-based metrics, particularly in the top half of the table. For example, CSC-32B, which is ranked 1st by the official test-based metric, drops to 4th place when evaluated by SPOTIT⁺. And the 3rd place method ALPHA drops to the 6th place. These results indicate that test-based methods can in many cases overlook differences between the generated SQL and the gold SQL, which might lead to misrepresentation of the actual performance (both *absolute* and *relative*) of existing Text-to-SQL methods.

**The effect of K.** To study the effect of the choice of the verification bound $K$, we vary its value from 1 to 7 and run SPOTIT on the predictions of CSC-32B, the best model according to EX-TEST. As shown in Fig. 4, SPOTIT was able to find significantly more differentiating databases when $K$ increases from 1 to 2 and the gain is marginal pass $K = 3$. This justifies our choice of $K = 5$.

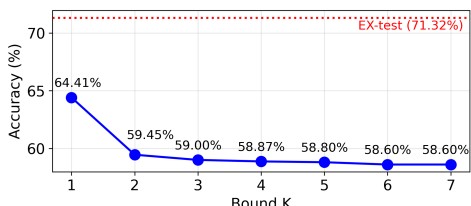

Figure 4: The effect of bound $K$.

**Manual inspection of SPOTIT counterexamples.** As SPOTIT performs bounded verification, the differentiating databases it finds are guaranteed to be minimal, which makes it easy to analyze them and understand the source of difference between the generated and gold SQLs. We manually examined the counterexamples for a random sample of 50 queries generated by CSC-32B and found that the difference between a generated SQL and a gold SQL can be primarily attributed to the three reasons that we described in Section 3: ambiguous question, incorrect gold SQL, and incorrect generated SQL. Fig. 5 shows a breakdown of the primary attributed reasons for those sampled questions. Surprisingly, while incorrect predictions do constitute a significant portion (26%), more often than not, the gold SQL itself is problematic. There are also a small fraction of cases (10%) where the question itself can be interpreted in multiple ways and therefore admits different answers. We discussed two

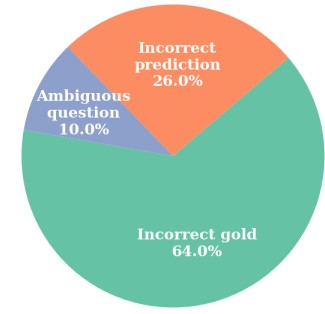

Figure 5: A breakdown of the primary reason for the difference between generated and gold SQLs.

examples of incorrect gold SQLs and ambiguous questions in Fig. 1. Additional examples of each type of issues, along with the databases found by SPOTIT, are provided and discussed in App. D.

**Additional analysis on Spider 2.0 benchmarks.**
To assess the generalizability of SPOTIT to more complex Text-to-SQL tasks, we evaluate it on the recently introduced Spider 2.0 benchmark (Lei et al., 2024). We consider OMNISQL (Li et al., 2025b), a state-of-the-art Text-to-SQL method and GPT-5 [2] on the 135 SQLite questions. These methods pass EX-TEST on 46 and 57 queries, re-

Table 4: Evaluation of SPOTIT on Spider 2.0.

| | EX-TEST | EX-SPOTIT | | |
|---|---|---|---|---|
| | Acc.(%) | Acc.(%) | Supported (%) | Avg. Time |
| OMNISQL | 34.1 | 22.2 | 60.9% | 3.4 |
| GPT-5 | 42.2 | 36.3 | 50.9% | 1.1 |

spectively, which is competitive with the top entries on the Spider 2.0 leaderboard.[3] As shown in Tab. 4, SPOTIT finds differentiating databases for 16 $((34.1\% - 22.2\%) * 135)$ and 8 query pairs deemed correct by test-based evaluation respectively for OMNISQL and GPT-5. SpotIt's runtime for finding counterexamples remains low. We believe this is due to the fact that the schemas in Spider 2.0 are only moderately larger than those in BIRD (97.6 vs. 78.6 columns) and counter-examples can usually be detected with small values of $K$. The main verification challenge when it comes to Spider 2.0 lies in the number of SQL operators required for the queries but currently unsupported by our verification engine, which resulted in a smaller percentage of supported query pairs. Upon closer examination, 52.6% of the unsupported Spider 2.0 queries involve the window function (i.e., OVER clauses). Overall, these results indicate that SPOTIT is also useful for uncovering query discrepancies overlooked by test-based methods on the challenging Spider 2.0 benchmarks.

**Summary of findings and implications.** We now summarize the findings of our evaluation of a state-of-the-art Text-to-SQL evaluation dataset BIRD using SPOTIT and discuss their implications.

*Finding 1: Existing test-based correctness metrics that involve executing the generated SQL and the gold SQL on static test databases can overlook significant variations in output data returned by the generated and gold SQLs.* A search-based evaluation metric, such as SPOTIT, can serve as a practical alternative that provides additional perspectives on the performance of Text-to-SQL methods.

*Finding 2: there is a significant number of problematic gold SQLs in existing Text-to-SQL benchmark sets.* As shown by examples in Tab. 1 and App. D, in many cases, the issue can be hard to detect, yet can cause significantly different behaviors from the intended one. The presence of incorrect gold SQLs makes it hard to determine the true optimal performance on a benchmark set, as even a perfect Text-to-SQL method cannot achieve 100% accuracy.

Based on our result analysis for CSC-32B, we speculate that when most Text-to-SQL methods disagree with the gold SQL, the gold SQL is likely problematic. To validate this, we count the number of times that a prediction for a question is deemed correct by EX-TEST but incorrect by SPOTIT[+] across *all* 10 Text-to-SQL methods. As shown in Fig. 6, there are 36 questions on which all methods generated queries that differ from the gold SQL. Manual inspection suggests that 31 of those 36 cases have problematic gold SQLs, 3 have ambiguous questions, and only 2 represent genuine errors in the generated SQLs.

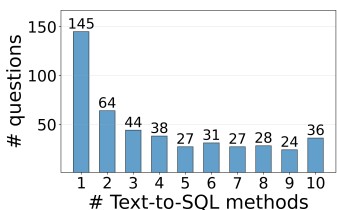

Figure 6: A breakdown of questions that passed EX-TEST but failed SPOTIT[+].

While so far we have focused on incorrect gold SQLs overlooked by EX-TEST, our investigation begs the question: *when the generated query differs from the gold SQL, how often in general is the gold SQL problematic?* Fig. 7 shows the number of times the prediction for a question is deemed incorrect by EX-TEST across the 10 Text-to-SQL methods, for questions where CSC-32B's predictions failed EX-TEST. There are 294 questions where at least 8 of the other 9 methods also failed EX-TEST. If shared disagreement with gold SQL is also a good indicator for problematic gold SQL in this case, then even a perfect Text-to-SQL method might

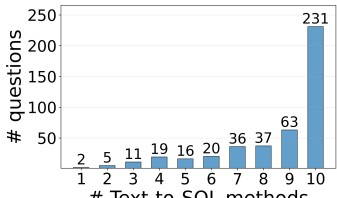

Figure 7: A breakdown of questions for which CSC-32B's predictions failed EX-TEST.

---

[2] We use the same prompt as used in OMNISQL.

[3] https://spider2-sql.github.io/. We were not able to obtain predictions of Text-to-SQL methods on the leaderboard because they are predominantly closed source.

not be able to achieve an EX-TEST score much higher than 80%
on BIRD-dev. As the time of completing this manuscript, the best EX-TEST score for BIRD-dev achieved by any method on the official leaderboard is 76.14%.

Large-scale benchmark sets inevitably contain problematic gold SQLs. Indeed, multiple sources have found examples of problematic gold SQLs in the BIRD dataset (Hui, 2024; Wretblad et al., 2024), and some of them have already been addressed by the maintainers. SPOTIT is the first approach that can provide minimal, easily analyzable databases to differentiate generated and gold SQLs, and can help to systematically uncover problematic gold SQLs.

*Finding 3: A substantial number of questions in the Text-to-SQL dataset can be interpreted in different ways, thus admitting different SQL queries.* While ambiguity is inherent in natural language, judging the correctness of a generated SQL query based on a single gold SQL query when the natural language question admits multiple interpretations might result in unfair penalization of Text-to-SQL methods.

*Finding 4 (for the verification community): SMT-based equivalence verification techniques can already support a large fraction of practical SQL queries.* Our results demonstrate that verification can often be completed within seconds. Due to the practical relevance of Text-to-SQL, we believe there is motivation for the verification community to invest more resources to precisely cover a larger fragment of SQL. In App. C, we discuss further extensions that would be especially useful according to our evaluation. Another significant next step to incorporate user preferences (potentially from natural languages) to search for particular types of counterexample databases. One way to achieve this is to encode the preferences as additional constraints in the SMT formulation of the verification problem.

## 6    RELATED WORK

Popular evaluation platforms such as BIRD-SQL (Li et al., 2024) and Spider 2.0 (Lei et al., 2024) evaluate query correctness by testing on predefined database instances. Several additional evaluations have been proposed to take into account partially correct generated queries (Pinna et al., 2025), efficiency of query executions (Zhang et al., 2024), and ambiguity in the questions (Li et al., 2023a; BIRD, 2025). However, the final correctness check is still via testing on a static database. Formal SQL equivalence checking broadly falls into two categories, full-fledged verification (Chu et al., 2017c; 2018; Zhou et al., 2022; 2024; Wang et al., 2024) and bounded verification (Veanes et al., 2010; Chu et al., 2017a;b; He et al., 2024). To the best of our knowledge, VERIEQL (He et al., 2024) supports the most expressive SQL fragments, while also offering extensibility for new features. We significantly extend the VERIEQL framework to support date and string types as well as a number of common operators for the Text-to-SQL evaluation task. Test data generation methods can also be useful for detecting query non-equivalence (Chandra et al., 2015; Somwase et al., 2024; Zhong et al., 2020).However, when the counterexamples require very specific structures, random fuzzing/testing can become unreliable in refuting equivalence. In contrast, SPOTIT systematically searches over the space of possible differentiating databases, finds minimal counter-examples, and provides a formal guarantee: if SPOTIT deems two SQL queries equivalent, then no counterexample of size less than a fixed number exists. A more detailed review of related work can be found in App. A.

## 7    CONCLUSION

We presented SPOTIT, the first verification-based evaluation pipeline for Text-to-SQL. We introduced techniques to support a richer SQL grammar, which enabled us to efficiently analyze a large fragment of SQL queries commonly seen in Text-to-SQL tasks. Our initial motivation for developing SPOTIT was to examine the extent to which the accuracy of a Text-to-SQL method is overestimated by test-based evaluation, which is widely adopted as the default metric on high-profile Text-to-SQL evaluation platforms. However, a closer inspection of the verification results revealed a far more complex picture. While SPOTIT can indeed detect incorrect generated SQL queries that were overlooked by test-based methods, a significant portion of the inconsistency between the gold and generated SQLs can be explained by the benchmarks themselves–either due to problematic gold SQLs or due to ambiguous natural language questions. We discussed the implications of and the next steps from our findings, and hope that our work will motivate further work on evaluating and improving Text-to-SQL evaluation frameworks.

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

# A  RELATED WORK

A large number of Text-to-SQL frameworks have been proposed over the last few years by research groups in academia and industry (Liu et al., 2023; Dong et al., 2023; Chang & Fosler-Lussier, 2023; TiDBCloud, 2020; Talaei et al., 2024b; Gao et al., 2025; Sequeda et al., 2023; Sheng & Xu, 2025b; Liu et al., 2025; Shkapenyuk et al., 2025; Zhai et al., 2025). However, evaluation frameworks have received much less attention. There are two main publicly available platforms: BIRD-SQL (Li et al., 2024) and Spider (Lei et al., 2024) that are commonly used to evaluate the performance of Text-to-SQL methods. Their evaluation procedure is performed on predefined database instances, whereas SPOTIT searches for a separation database instance. A number of evaluation metrics were proposed to take into account partially correct generated queries (Pinna et al., 2025) or the efficiency of query executions (Zhang et al., 2024). Recently, (Li et al., 2023a; BIRD, 2025) proposed an iterative evaluation framework in which the system can interact with the user by asking additional questions (e.g., to resolve ambiguity). However, the final evaluation of the correctness of the generated SQL query is still performed on a static database.

There are two lines of work in formal equivalence checking for SQL queries: full-fledged and bounded verification. The full-fledged methods (Chu et al., 2017c; 2018; Zhou et al., 2022; 2024; Wang et al., 2024) encode queries into specific representations (e.g., algebraic expressions (Chu et al., 2018; Wang et al., 2024)) and determine equivalence by proving the equivalence of these representations, thereby guaranteeing equivalence of queries for any possible database. However, such methods typically support only a limited subset of SQL and cannot generate counterexamples for non-equivalent queries. In contrast, the bounded verification approaches (Veanes et al., 2010; Chu et al., 2017a;b; He et al., 2024) check equivalence within a finite search space, making them capable of handling larger subsets of SQL and identifying counterexamples. To the best of our knowledge, VERIEQL supports the most expressive SQL fragments and rich integrity constraints, while also offering extensibility for new features (He et al., 2024). In this work, we significantly extend the VERIEQL framework to support date and string types as well as a number of common operators for the Text-to-SQL evaluation task.

Test data generation methods can also be useful for detecting query non-equivalence (Chandra et al., 2015; Somwase et al., 2024; Zhong et al., 2020).However, when the counterexamples require very specific structures (which is the case for many query pairs that passed EX-test but failed SpotIt as seen in App. D), random fuzzing/testing can become unreliable in refuting equivalence. In contrast, SPOTIT systematically searches over the space of possible differentiating databases, finds minimal counter-examples, and provides a formal guarantee: if two SQL queries are considered equivalent, then no counterexample of size less than a fixed number exists. Computational resources permitted, one could in principle run a portfolio of test-based and verification-based equivalence-checking methods in parallel to more quickly detect non-equivalence. This is an orthogonal but interesting future direction.

## B  ANALYSIS OF RUNTIME

Tables 5, 6, 7, 8, and 9 show the effect of different parameters on runtime, including the number of columns, integrity constraints, tables in the databases, the number of sub-queries in the gold SQL, and the number of nodes in the abstract syntax tree in the gold SQL. We found that all parameters except for the number of tables are positively correlated with median runtime.

| #columns | Median runtime (s) |
|---|---|
| 11 | 0.4310 |
| 21 | 0.1701 |
| 31 | 0.1963 |
| 48 | 0.3887 |
| 55 | 0.5576 |
| 64 | 0.3007 |
| 71 | 0.5365 |
| 89 | 0.2672 |
| 94 | 0.3154 |
| 115 | 0.6537 |
| 199 | 0.8006 |

Table 5: Median runtime by number of columns

| #constraints | Median runtime (s) |
|---|---|
| 5 | 0.2842 |
| 7 | 0.1701 |
| 10 | 0.4324 |
| 16 | 0.5576 |
| 17 | 0.3887 |
| 19 | 0.1963 |
| 21 | 0.5365 |
| 36 | 0.6877 |

Table 6: Median runtime by number of constraints

| #tables | Median runtime (s) |
|---|---|
| 3 | 0.2842 |
| 4 | 0.4310 |
| 5 | 0.1701 |
| 6 | 0.6537 |
| 7 | 0.8006 |
| 8 | 0.4249 |
| 10 | 0.1963 |
| 13 | 0.3154 |

Table 7: Median runtime by number of tables

| #subqueries | Median runtime (s) |
|---|---|
| 0 | 0.3676 |
| 1 | 0.3972 |
| 2 | 1.9128 |

Table 8: Median runtime by number of subqueries

| #AST nodes | Median runtime (s) |
|---|---|
| 0–19 | 0.1812 |
| 20–39 | 0.3408 |
| 40–59 | 0.3426 |
| 60–79 | 0.4216 |
| 80–99 | 0.9683 |
| 100–119 | 1.1865 |
| 120–139 | 0.2053 |
| 140–159 | 1.2395 |

Table 9: Median runtime by number of AST nodes (buckets of 20)

## C  LIMITATIONS OF EXISTING BOUNDED SQL EQUIVALENCE CHECKER

While SPOTIT builds on top of and extends VERIEQL, a state-of-the-art bounded verifier that claims to cover the largest SQL fragment, we find that there are still SQL operators which it either does not support or cannot precisely capture. In this section, we describe the features that appear frequently in failure cases of SPOTIT.

- The gold query for question 726 in the BIRD-dev benchmark.

      SELECT superhero_name, height_cm,
              RANK() OVER (ORDER BY height_cm DESC) AS HeightRank
      FROM superhero INNER JOIN publisher
              ON superhero.publisher_id = publisher.id
      WHERE publisher.publisher_name = "Marvel Comics"

  SPOTIT does not support the window and analytic functions such as RANK and LAG.

- A SQL query generated by OMNISQL.

      WITH RECURSIVE TimeSeriesAS (
              SELECT '2016-01-01' AS mth
              UNION ALL
              SELECT DATE(mth, '+1 month') AS mth FROM TimeSeries
              WHERE mth < '2017-12-01'
          ),
          . . .
      SELECT product_name FROM SalesRatio . . . ORDER BY product_name

  SPOTIT cannot encode recursive common table expressions above.

- Imprecisely encoding for ORDER BY and LIMIT:

  Since VERIEQL establishs tables under bag semantics (namely, multi-set), database instances are considered equivalent if they has the same tuples with the same multiplicities. For instance, $T_1$ and $T_2$ in Tables 10–11 are equivalent under bag semantics.

  However, when VERIEQL symbolically execute ORDER BY A, VERIEQL automatically converts semantics from bag to list in which the order of tuples matters. In such a case, database instances is equivalent iff they are tuple-wise the same. For instance, $T_1$ and $T_2$ in Tables 10–11 are not equivalent under list semantics. Furthermore, if ORDER BY A is followed by LIMIT 1, then execution results on $T_1$ and $T_2$ are, respectively, $R_1$ and $R_2$.

Table 10: $T_1$

|  | A | B |
|---|---|---|
| $R_1$ | 1 | 2 |
| $R_2$ | 1 | 3 |

Table 11: $T_2$

|  | A | B |
|---|---|---|
| $R_2$ | 1 | 3 |
| $R_1$ | 1 | 2 |

More concretely, consider the gold query $Q_1$ of question 653 in the BIRD-dev benchmark, a query $Q_2$ generated by ALPHA, and a counterexample database found by VERIEQL as follows:

The gold query $Q_1$:

SELECT DisplayName FROM users WHERE id = (
     SELECT OwnerUserId FROM posts ORDER BY ViewCount DESC LIMIT 1
)

The generated query $Q_2$:

     SELECT u.displayname AS ownerdisplayname
     FROM posts AS p INNER JOIN users AS u ON p.owneruserid = u.id
     ORDER BY p.viewcount DESC LIMIT 1

Table 12: *posts*

| viewcount | owneruserid |
|-----------|-------------|
| Null | 1 |
| Null | 0 |

Table 13: *users*

| id | displayname |
|----|-------------|
| 0 | 'A' |
| 1 | 'B' |

VERIEQL's execution results of $Q_1$ and $Q_2$ are shown in Tables 14–15.

Table 14: VERIEQL's result of $Q_1$

| DisplayName |
|-------------|
| 'A' |

Table 15: VERIEQL's result of $Q_2$

| ownerdisplayname |
|------------------|
| 'B' |

SQLite's execution results of $Q_1$ and $Q_2$ are shown in Tables 16–17.

Table 16: SQLite's result of $Q_1$

| DisplayName |
|-------------|
| 'A' |

Table 17: SQLite's result of $Q_2$

| ownerdisplayname |
|------------------|
| 'A' |

Naturally, VERIEQL identifies a spurious counterexample where $Q_1$'s result is 'B' instead of 'A'. This is because the intermediate table from the FROM clause of $Q_2$ is shown as Table 18 where the values in column "viewcount" are all Null values. While executing the ORDER BY, VERIEQL does not reorder the tuples of this intermediate table but SQLite engine will swap these two tuples. Therefore, VERIEQL failed in this verification task.

Table 18: VERIEQL's intermediate table of $Q_2$

| u.DisplayName | p.viewcount |
|---------------|-------------|
| 'A' | Null |
| 'B' | Null |

# D  ADDITIONAL INCONSISTENCY BETWEEN PREDICTED AND GOLD SQLS OVERLOOKED BY EX-TEST

## D.1  EXAMPLE 3.1 (EXTENDED)

Tables 19–20 show a counterexample database $D_{\text{cex}}$ (these are two relevant tables). The generated SQL $P$ returns no records, since `laboratory.rnp` is equal to '+-' in the single record that violated `NOT laboratory.rnp IN ('-', '+-')`. In contrast, the gold SQL $Q$ returns '1000-01-01', because the condition `T2.rnp != '-' OR '+-'` is incorrect.

### Table 19: `patient`

| id | sex | birthday | description | first_date | admission | diagnosis |
|----|-----|----------|-------------|------------|-----------|-----------|
| 0 | '1' | '1000-01-01' | '1000-01-01' | '1000-01-01' | '1' | '1' |

### Table 20: `laboratory` (skipped irrelevant columns)

| id | date | got | gpt | ldh | RNP | ... |
|----|------|-----|-----|-----|-----|-----|
| 0 | '1000-01-01' | 0 | 0 | 0 | '+-' | ... |

## D.2  EXAMPLE 3.2 (EXTENDED)

Tables 21–22 show a counterexample database $D_{\text{cex}}$ (these are two relevant tables).

The generated SQL $P$ counts two records while the gold SQL $Q$ counts only one record, because the `DISTINCT` operator is applied before counting.

### Table 21: `examination` (skipped irrelevant columns)

| id | examination_date | acl_igg | acl_igm | ana | ana_pattern | acl_iga | diagnosis | kct | rvvt | lac | ... |
|----|------------------|---------|---------|-----|-------------|---------|-----------|-----|------|-----|-----|
| 1 | '1000-01-01' | 11 | 12 | 0 | '1' | 0 | '1' | '1' | '1' | '1' | ... |
| 1 | '1000-01-01' | 14 | 15 | 0 | '1' | 0 | '1' | '1' | '1' | '1' | ... |

### Table 22: `patient`

| id | sex | birthday | description | first_date | admission | diagnosis |
|----|-----|----------|-------------|------------|-----------|-----------|
| 0 | '1' | '1000-01-01' | '1000-01-01' | '1000-01-01' | '1' | '1' |
| 1 | '1' | '1000-01-01' | '1000-01-01' | '1000-01-01' | '1' | '1' |

## D.3 ADDITIONAL EXAMPLES

**Example D.1.** *Consider the question $N_3$ and the corresponding SQL queries (Figure 8). The differentiating database found by* SPOTIT *is shown in Tables 23, 24, 25. Note that there is a typo in the evidence. According to external medical sources, the normal range of uric acid levels in females should be defined as less than or equal to 6.50, not greater than. The annotator overlooked this typo, and as a result, the gold SQL is clearly incorrect.* □

```
N3: "What is the anti Cardiolipin antibody concentration of the female patient
with the highest uric acid level in the normal range?"
Evidence: "Anti Cardiolipin antibody concentration refers to 'aCL IgG', 'aCL IgM', 'aCL IgA';
female patient refers to Sex = F'; highest uric acid level in the normal range refers to MAX(UA > 6.50);"

/*Gold SQL Q*/:
SELECT T3.acl_igg, T3.acl_igm, T3.acl_iga
FROM patient AS T1
INNER JOIN laboratory AS T2 ON T1.id = T2.id
INNER JOIN examination AS T3 ON T3.id = T2.id
WHERE T1.sex = 'F' AND  T2.ua > 6.5
ORDER BY T2.ua DESC
LIMIT 1

/*Generated SQL P*/:
SELECT examination.acl_igg, examination.acl_igm, examination.acl_iga
FROM patient
INNER JOIN laboratory ON patient.id = laboratory.id
INNER JOIN examination ON patient.id = examination.id
WHERE patient.sex = 'F' AND  laboratory.ua <= 6.5
ORDER BY laboratory.ua DESC
LIMIT 1
```

Figure 8: An example of a query with an incorrect gold SQL.

Table 23: `patient` (skipped irrelevant columns)

| id | sex | ... |
|----|-----|-----|
| 0  | 'F' | ... |

Table 24: `laboratory` (skipped irrelevant columns)

| id | ua | ... |
|----|-----|-----|
| 0  | 6.5 | ... |

Table 25: `examination` (skipped irrelevant columns)

| id | acl_igg | acl_igm | acl_iga | ... |
|----|---------|---------|---------|-----|
| 0  | 1       | 1       | 1       | ... |

***Example* D.2.** *Consider the question $N_4$ and the corresponding SQL queries (Figure 9). The differentiating database found by* SPOTIT *is shown in Tables 26,27. The natural langue question asks for transactions after January 1st, 2012, which requires excluding January 1st, 2012. However, the gold SQL uses a greater-than-or-equal-to condition, which includes 2012/01/01, thus being incorrect.* □

---

$N_4$: *"Among the transactions made in gas stations in the Czech Republic, how many took place after 2012/1/1?"*
*Evidence: "Country code for Czech Republic is 'CZE'."*

```
/*Gold SQL Q*/:
SELECT COUNT(T1.transactionid)
FROM transactions_1k AS T1
INNER JOIN gasstations AS T2 ON T1.gasstationid = T2.gasstationid
WHERE T2.country = 'CZE' AND STRFTIME('%Y', T1.date) ≥ '2012';

/*Generated SQL P*/:
SELECT COUNT(*)
FROM transactions_1k AS T
INNER JOIN gasstations AS G ON T.gasstationid = G.gasstationid
WHERE G.country = 'CZE' AND T.date > '2012-01-01';
```

Figure 9: An example of a query with an incorrect gold SQL.

Table 26: `transactions_1k` (skipped irrelevant columns)

| transaction_id | gasstation_id | date | ... |
|---|---|---|---|
| 0 | 0 | '2012-01-01' | ... |

Table 27: `gasstations` (skipped irrelevant columns)

| gasstation_id | country | ... |
|---|---|---|
| 0 | 'CZE' | ... |

**Example** D.3. *Consider the question $N_5$ and the corresponding SQL queries (Figure 10). The differentiating database found by* SPOTIT *is shown in Tables 28,29. This example demonstrates an incorrect gold SQL, which orders by the latest time (DESC) rather than the earlier time (ASC). This directly contradicts the natural language question.* ☐

---

$N_5$: *"Which country's gas station had the first paid customer in 2012/8/25?"*
Evidence: *"2012/8/25' can be represented by '2012-08-25'."*

```
/*Gold SQL Q*/:
SELECT T2.country
FROM transactions_1k AS T1
INNER JOIN gasstations AS T2 ON T1.gasstationid = T2.gasstationid
WHERE T1.date = '2012-08-25'
ORDER BY  T1.time DESC
LIMIT 1;
/*Generated SQL P*/:
SELECT G.country
FROM gasstations AS G
JOIN (
  SELECT gasstationid
  FROM transactions_1k
  WHERE date = '2012-08-25'
  ORDER BY  time ASC LIMIT 1
) AS T
ON G.gasstationid = T.gasstationid;
```

Figure 10: An example of a query with an incorrect gold SQL.

Table 28: transactions_1k (skipped irrelevant columns)

| gasstation_id | date | time | ... |
|---|---|---|---|
| 0 | '2012-08-25' | 1 | ... |
| 0 | '2012-08-25' | 2 | ... |

Table 29: gasstations (skipped irrelevant columns)

| gasstation_id | country | ... |
|---|---|---|
| 0 | '1' | ... |

**Example D.4.** *Consider the question $N_6$ and the corresponding SQL queries (Figure 11). The differentiating database found by* SPOTIT *is shown in Tables 30, 31. The gold SQL incorrectly encodes the exclusive inequality specified in the natural langue question by using the BETWEEN operator, which leads to inclusive bounds. Thus, the gold SQL is incorrect as it includes values outside of the specified range.* □

---

$N_6$: *"Please list a patient's platelet level if it is within the normal range*
*and if he or she is daignosed with MCTD"*
*Evidence: "PLT > 100 and PLT < 400 means platelet level is within the normal range;*
*PLT < 100 and PLT > 400 means platelet level is not within the normal range;*
*diagnosed with MCTD refers to Diagnosis = 'MCTD'";*

```
/*Gold SQL Q*/:
SELECT T2.plt
FROM patient AS T1
INNER JOIN laboratory AS T2 ON T1.id = T2.id
WHERE T1.diagnosis = 'MCTD' AND  T2.plt BETWEEN 100 AND 400
/*Generated SQL P*/:
SELECT L.plt
FROM LABORATORY L
INNER JOIN PATIENT P ON L.id = P.id
WHERE P.diagnosis = 'MCTD' AND  L.plt > 100 AND L.plt < 400
```

Figure 11: An example of a query with an incorrect gold SQL.

Table 30: `patient` (skipped irrelevant columns)

| id | diagnosis | ... |
|----|-----------|-----|
| 0  | 'MCTD'    | ... |

Table 31: `laboratory` (skipped irrelevant columns)

| id | plt | ... |
|----|-----|-----|
| 0  | 100 | ... |

**Example** **D.5.** *Consider the question $N_7$ and the corresponding SQL queries (Figure 12). The differentiating database found by* SPOTIT *is shown in Tables 32, 33. In this example, the generated SQL is incorrect as it is clearly missing the link_to_major constraint, filtering only by name.* ☐

---

$N_7$: *"Please indicate the college of the person whose first name is Katy with the link to the major 'rec1N0upiVLy5esTO' "*

```
/*Gold SQL Q*/:
SELECT T2.college
FROM member AS T1
INNER JOIN major AS T2 ON T2.major_id = T1.link_to_major
WHERE   T1.link_to_major = 'rec1N0upiVLy5esTO' AND T1.first_name = 'Katy'
/*Generated SQL P*/:
SELECT major.college
FROM member
INNER JOIN MAJOR ON member.link_to_major = major.major_id
WHERE   member.first_name = 'Katy'
```

Figure 12: An example of a query with an incorrect generated SQL.

Table 32: member (skipped irrelevant columns)

| link_to_major | first_name | ... |
|---|---|---|
| '1' | 'Katy' | ... |

Table 33: major (skipped irrelevant columns)

| major_id | college | ... |
|---|---|---|
| 1 | '0' | ... |

*Example* **D.6.** *Consider the question $N_8$ and the corresponding SQL queries (Figure 13). The differentiating database found by* SPOTIT *is shown in Tables 34, 35. In this example, the generated SQL only checks whether the patient was diagnosed with SLE on January 1st, 1997. However, the natural language question also asks for the patient's original diagnose at their first hospital visit. Since the generated SQL doesn't include this condition, it's incorrect as it could return a diagnoses from a later visit rather than the patient's first one.* □

---

$N_8$: *"For the patient who was diagnosed SLE on 1997/1/27, what was his/her original diagnose when he/she came to the hospital for the first time?"*

*Evidence: "'SLE' and original diagnose refers to Diagnosis; 1997/1/27 refers to 'Examination Date' = '1997-01-27'; first came to the hospital refers to patient.'First Date'."*

```
/*Gold SQL Q*/:
SELECT T1.diagnosis
FROM patient AS T1
INNER JOIN examination AS T2 ON T1.id = T2.id
WHERE T1.id = (
  SELECT id
  FROM examination
  WHERE examination_date = '1997-01-27' AND diagnosis = 'SLE'
) AND T2.examination_date = T1.first_date;

/*Generated SQL P*/:
SELECT T2.diagnosis
FROM examination AS T1
INNER JOIN patient AS T2 ON T1.id = T2.id
WHERE T1.diagnosis = 'SLE' AND T1.examination_date = '1997-01-27';
```

Figure 13: An example of a query with an incorrect generated SQL.

Table 34: `patient` (skipped irrelevant columns)

| id | diagnosis | first_date | ... |
|---|---|---|---|
| 0 | '1' | '1997-01-26' | ... |

Table 35: `examination` (skipped irrelevant columns)

| id | examination_date | diagnosis | ... |
|---|---|---|---|
| 0 | '1997-01-27' | 'SLE' | ... |

***Example* D.7.** *Consider the question $N_9$ and the corresponding SQL queries (Figure 14). The differentiating database found by* SPOTIT *is shown in Tables 36,37. This is an example of an ambiguous question. The term 'members' can be interpreted in at least two ways: any student who is a part of the club, or more specifically, students in the club with the recorded position of 'member'. While the gold SQL takes the second interpretation, filtering on T2.position = 'Member', it's just as reasonable to assume that all students in the club are members, and leave out a secondary filter. Coupled with the lack of evidence, the resulting difference in queries is most likely due to the ambiguity of the natural language question. Hence, it's been marked as an ambiguous question.* □

---

$N_9$*: "List the last name of members with a major in environmental engineering and include its department and college name.*
*Evidence: "Environmental Engineering' is the major name"*

```
/*Gold SQL Q*/:
SELECT T2.last_name, T1.department, T1.college
FROM major AS T1
INNER JOIN member AS T2 ON T1.major_id = T2.link_to_major
WHERE T2.position = 'Member' AND T1.major_name = 'Enviormental Engineering'
/*Generated SQL P*/:
SELECT T1.last_name, T2.department, T2.college
FROM member AS T1
INNER JOIN major AS T2 ON T1.link_to_major = T2.major_id
WHERE T2.major_name = 'Enviormental Engineering'
```

Figure 14: An example of an ambiguous question.

Table 36: major (skipped irrelevant columns)

| major_id | major_name | department | college | ... |
|---|---|---|---|---|
| 0 | 'Environmental Engineering' | '1' | '1' | ... |

Table 37: member (skipped irrelevant columns)

| last_name | link_to_major | position | ... |
|---|---|---|---|
| '1' | 0 | '1' | ... |

**Example D.8.** *Consider the question $N_{10}$ and the corresponding SQL queries (Figure 15). The differentiating database found by SPOTIT is shown in Tables 38, 39. This example is marked as ambiguous because the natural language question is underspecified. If the intent is to return the legal status of every valid artifact card, which is a reasonable interpretation, than the generated SQL would be correct. However, if the intent is to return the set of unique legal statuses across valid artifact cards, than the gold SQL is correct.* □

---

*$N_{10}$: "For artifact type of cards that do not have multiple faces on the same card, state its legalities status for vintage play format."*
*Evidence: "Artifact type of cards refers to types = 'Artifact'; card does not have multiple faces on the same card refers to side is NULL'; vintage play format refers to format = 'vintage';"*

```
/*Gold SQL Q*/:
SELECT  DISTINCT T2.status
FROM cards AS T1
INNER JOIN legalities AS T2 ON T1.uuid = T2.uuid
WHERE T1.type = 'Artifact' AND T2.format = 'vintage' AND T1.side IS NULL;

/*Generated SQL P*/:
SELECT  T2.status
FROM cards AS T1
JOIN legalities AS T2 ON T1.uuid = T2.uuid
WHERE T1.type = 'Artifact' AND T1.side IS NULL AND T2.format = 'vintage';
```

---

Figure 15: An example of an ambiguous question.

Table 38: cards (skipped irrelevant columns)

| uuid | type | side | ... |
|------|------|------|-----|
| '0' | 'Artifact' | NULL | ... |

Table 39: legalities (skipped irrelevant columns)

| uuid | format | status | ... |
|------|--------|--------|-----|
| '0' | 'vintage' | '1' | ... |
| '0' | 'vintage' | '1' | ... |

**Example** **D.9.** *Consider the question $N_{11}$ and the corresponding SQL queries (Figure 16). The differentiating database found by* SPOTIT *is shown in Tables 40, 41. This example is considered ambiguous because the natural language question and evidence do not specify a tie-breaking rule. In the case that there are two comments on valid posts with a tied high score, a query with LIMIT 1 may return either comment. This is why the generated and gold SQL return different results. Since the difference arises solely from a lack of specificity, this example is marked as ambiguous.* □

---

$N_{11}$: *"Among the posts with views ranging from 100 to 150, what is the comment with the highest score?"*
*Evidence: "Views ranging from 100 to 150 refers to ViewCount BETWEEN 100 and 150; comment with the highest score refers to Text where MAX(Score);"*

---

```
/*Gold SQL Q*/:
SELECT text
FROM comments
WHERE postId IN (
  SELECT id
  FROM posts
  WHERE viewCount BETWEEN 100 AND 150
) ORDER BY score DESC
LIMIT 1

/*Generated SQL P*/:
SELECT T2.text
FROM posts AS T1
INNER JOIN comments AS T2 ON T1.id = T2.postId
WHERE T1.viewCount BETWEEN 100 AND 150
ORDER BY T2.score DESC
LIMIT 1
```

---

Figure 16: An example of an ambiguous question.

Table 40: `comments` (skipped irrelevant columns)

| postId | score | text | ... |
|--------|-------|------|-----|
| 0 | 1 | '1' | ... |
| 1 | 1 | '2' | ... |

Table 41: `posts` (skipped irrelevant columns)

| id | viewCount | ... |
|----|-----------|-----|
| 0 | 100 | ... |
| 1 | 100 | ... |

## E  SEMANTICS

$$\llbracket E \rrbracket :: \text{Database } D \to \text{Relation} \to \text{Value}$$

$\llbracket \text{ToInt}(E) \rrbracket_{D,xs}$ $=$ let $v = \llbracket E \rrbracket_{D,xs}$ in
  $\text{ite}(v = \text{Null} \lor \text{IsInt}(v), v,$
    $\text{ite}(\text{IsStr}(v), \llbracket \text{StrToInt}(v) \rrbracket_{D,xs}, \llbracket \text{DateToInt}(v) \rrbracket_{D,xs}))$

$\llbracket \text{ToDate}(E) \rrbracket_{D,xs}$ $=$ let $v = \llbracket E \rrbracket_{D,xs}$ in
  $\text{ite}(v = \text{Null} \lor \text{IsDate}(v), v,$
    $\text{ite}(\text{IsInt}(v), \llbracket \text{IntToDate}(v) \rrbracket_{D,xs}, \llbracket \text{StrToDate}(v) \rrbracket_{D,xs}))$

$\llbracket \text{ToStr}(E) \rrbracket_{D,xs}$ $=$ let $v = \llbracket E \rrbracket_{D,xs}$ in
  $\text{ite}(v = \text{Null} \lor \text{IsStr}(v), v,$
    $\text{ite}(\text{IsInt}(v), \llbracket \text{IntToStr}(v) \rrbracket_{D,xs}, \llbracket \text{DateToStr}(v) \rrbracket_{D,xs}))$

$\llbracket \text{DateToInt}(vs) \rrbracket_{D,xs}$ $=$ $\text{ite}(vs = \text{Null}, \text{Null}, vs[0] * 10^4 + vs[1] * 10^2 + vs[2])$

$\llbracket \text{StrToInt}(s) \rrbracket_{D,xs}$ $=$ let
  $v = \text{ite}(\text{IsDigits}(s), \text{StrToInt}(s),$
    $\text{ite}(s[0] = \text{“-”} \land \text{IsDigits}(s[1:]), -\text{StrToInt}(s), 0))$
  in $\text{ite}(s = \text{Null}, \text{Null}, v)$

$\llbracket \text{IntToStr}(v) \rrbracket_{D,xs}$ $=$ $\text{ite}(v = \text{Null}, \text{Null}, \text{IntToStr}(v))$

$\llbracket \text{DateToStr}(vs) \rrbracket_{D,xs}$ $=$ let
  $y = \text{IntToStr}(vs[0]),$
  $m = \text{ite}(vs[1] \leq 9, \text{“0”} + \text{IntToStr}(vs[1]),$
    $\text{IntToStr}(vs[1])),$
  $d = \text{ite}(vs[2] \leq 9, \text{“0”} + \text{IntToStr}(vs[2]),$
    $\text{IntToStr}(vs[2]))$
  in $\text{ite}(vs = \text{Null}, \text{Null}, y + \text{“-”} + m + \text{“-”} + d)$

$\llbracket \text{IntToDate}(v) \rrbracket_{D,xs}$ $=$ let $v_1 = \lfloor v/10^4 \rfloor, v_2 = \lfloor (v\%10^4)/10^2 \rfloor, v_3 = v\%10^2$ in
  $\text{ite}(v = \text{Null} \lor \text{IsValidDate}(v), \text{Null}, [v_1, v_2, v_3])$

$\llbracket \text{StrToDate}(s) \rrbracket_{D,xs}$ $=$ let $v = \llbracket \text{StrToInt}(s) \rrbracket_{D,xs}$ in
  $\text{ite}(s = \text{Null}, \text{Null}, \llbracket \text{IntToDate}(v) \rrbracket_{D,xs})$

$\llbracket E_1 \diamond E_2 \rrbracket_{D,xs}$ $=$ let
  $v_1 = \llbracket \text{ToInt}(E_1) \rrbracket_{D,xs}$ and $v_2 = \llbracket \text{ToInt}(E_2) \rrbracket_{D,xs},$
  in $\text{ite}(v_1 = \text{Null} \lor v_2 = \text{Null}, \text{Null}, v_1 \diamond v_2)$

$\llbracket \text{SubStr}(E_1, E_2, E_3) \rrbracket_{D,xs}$ $=$ let
  $e_i = \llbracket E_i \rrbracket_{D,xs}, e'_1 = \llbracket \text{ToStr}(e_1) \rrbracket_{D,xs}, l = \text{len}(e'_1),$
  $e'_2 = \llbracket \text{ToInt}(e_2) \rrbracket_{D,xs}, e'_3 = \llbracket \text{ToInt}(e_3) \rrbracket_{D,xs},$
  $v = \text{ite}(-l \leq e'_2 < 0, e_2 + l, \text{ite}(0 < e'_2 \leq l, e'_2 - 1, l + 1)),$
  $s = \text{ite}(v = 0 \lor v < -l \lor v > l \lor e'_3 \leq 0, \varepsilon,$
    $\text{ite}(e'_3 \geq l - v, e'_1[v:l], e'_1[v:2v + e'_3]))$
  in $\text{ite}(e_1 = \text{Null} \lor \text{IsStr}(e_2) \lor \text{IsStr}(e_3), \text{Null}, s)$

$\llbracket \text{Concate}(E_1, E_2) \rrbracket_{D,xs}$ $=$ let $v_1 = \llbracket \text{ToStr}(E_1) \rrbracket_{D,xs}$ and $v_2 = \llbracket \text{ToStr}(E_2) \rrbracket_{D,xs}$ in
  $\text{ite}(v_1 = \text{Null} \lor v_2 = \text{Null}, \text{Null}, \text{Concat}(v_1, v_2))$

$\llbracket \text{Strftime}(\kappa, E) \rrbracket_{D,xs}$ $=$ let $v = \llbracket \text{ToDate}(E) \rrbracket_{D,xs}$ in
  $\text{ite}(\kappa = \text{“\%Y”}, v[0], \text{ite}(\kappa = \text{“\%M”}, v[1], v[2]))$

$\llbracket \text{JulianDay}(E) \rrbracket_{D,xs}$ $=$ let $v = \llbracket E \rrbracket_{D,xs}$ in $\text{ToJulianDay}(v), \text{ if } \text{IsDate}(v)$

$\llbracket \text{DateShift}(E, i, \delta) \rrbracket_{D,xs}$ $=$ let $v = \llbracket E \rrbracket_{D,xs}$ in $\text{DateAdd}(v, i, \delta), \text{ if } \text{IsDate}(v)$

$$\llbracket \phi \rrbracket :: \text{Database } D \to \text{Relation} \to \text{Bool} \cup \text{Null}$$

$\llbracket \text{PrefixOf}(s, E) \rrbracket_{D,xs}$ $=$ let $v = \llbracket \text{ToStr}(E) \rrbracket_{D,xs}$ in $\text{ite}(v = \text{Null}, \text{Null}, \text{PrefixOf}(s, v))$

$\llbracket \text{SuffixOf}(s, E) \rrbracket_{D,xs}$ $=$ let $v = \llbracket \text{ToStr}(E) \rrbracket_{D,xs}$ in $\text{ite}(v = \text{Null}, \text{Null}, \text{SuffixOf}(s, v))$

$\llbracket \text{Like}(s, E) \rrbracket_{D,xs}$ $=$ let $v = \llbracket \text{ToStr}(E) \rrbracket_{D,xs}$ in $\text{ite}(v = \text{Null}, \text{Null}, \text{RegexMatch}(s, v))$

$\llbracket \text{Contain}(s, E) \rrbracket_{D,xs}$ $=$ let $s' = \text{Concat}(\text{“\%”}, s, \text{“\%”})$ in $\llbracket \text{Like}(s', E) \rrbracket_{D,xs}$

$\llbracket E_1 \odot E_2 \rrbracket_{D,xs}$ $=$ let $v_1 = \llbracket E_1 \rrbracket_{D,xs}$ and $v_2 = \llbracket E_2 \rrbracket_{D,xs}$ in
  $\text{ite}(v_1 = \text{Null} \lor v_2 = \text{Null}, \bot, v_1 \odot v_2), \text{ if } \text{Type}(v_1) = \text{Type}(v_2)$

Figure 17: Formal semantics for extended expressions and predicates. The IsValidDate function checks whether a string represent a date within the supported date range of a database engine. The ToJulianDay function converts a date to a Julian day and the DateAdd function move a date-value by modifier arguments $i$ and $\delta$. The definition of these two functions are shown in Appendix G.

# F ENCODING

$$\llbracket \text{ToInt}(E) \rrbracket_{\mathcal{S},\Gamma,\mathcal{T}} = \text{let } v = \llbracket E \rrbracket_{\mathcal{S},\Gamma,\mathcal{T}} \text{ in}$$
$$\text{ite}(v = \text{Null} \vee \text{IsInt}(v), v,$$
$$\text{ite}(\text{IsStr}(v), \llbracket \text{StrToInt}(v) \rrbracket_{\mathcal{S},\Gamma,\mathcal{T}}, \llbracket \text{DateToInt}(v) \rrbracket_{\mathcal{S},\Gamma,\mathcal{T}}))$$

$$\llbracket \text{ToDate}(E) \rrbracket_{\mathcal{S},\Gamma,\mathcal{T}} = \text{let } v = \llbracket E \rrbracket_{\mathcal{S},\Gamma,\mathcal{T}} \text{ in}$$
$$\text{ite}(v = \text{Null} \vee \text{IsDate}(v), v,$$
$$\text{ite}(\text{IsInt}(v), \llbracket \text{IntToDate}(v) \rrbracket_{\mathcal{S},\Gamma,\mathcal{T}}, \llbracket \text{StrToDate}(v) \rrbracket_{\mathcal{S},\Gamma,\mathcal{T}}))$$

$$\llbracket \text{ToStr}(E) \rrbracket_{\mathcal{S},\Gamma,\mathcal{T}} = \text{let } v = \llbracket E \rrbracket_{\mathcal{S},\Gamma,\mathcal{T}} \text{ in}$$
$$\text{ite}(v = \text{Null} \vee \text{IsStr}(v), v,$$
$$\text{ite}(\text{IsInt}(v), \llbracket \text{IntToStr}(v) \rrbracket_{\mathcal{S},\Gamma,\mathcal{T}}, \llbracket \text{DateToStr}(v) \rrbracket_{\mathcal{S},\Gamma,\mathcal{T}}))$$

$$\llbracket \text{DateToInt}(vs) \rrbracket_{\mathcal{S},\Gamma,\mathcal{T}} = \text{ite}(vs = \text{Null}, \text{Null}, vs[0] * 10^4 + vs[1] * 10^2 + vs[2])$$

$$\llbracket \text{StrToInt}(s) \rrbracket_{\mathcal{S},\Gamma,\mathcal{T}} = \text{let}$$
$$s_1 = s[1 : \text{z3.Length}(s)], v_1 = \text{z3.StrToInt}(s_1),$$
$$v = \text{ite}(s[0] = \text{``-''}, -v_1, \text{z3.StrToInt}(s)),$$
$$\Phi = \text{ite}(v < 0, \text{z3.IntToStr}(-v) = v_1, \text{z3.IntToStr}(v) = s),$$
$$\text{in ite}(s = \text{Null}, \text{Null}, \text{ite}(\Phi, v, 0))$$

$$\llbracket \text{IntToStr}(v) \rrbracket_{\mathcal{S},\Gamma,\mathcal{T}} = \text{ite}(v = \text{Null}, \text{Null}, \text{z3.IntToStr}(v))$$

$$\llbracket \text{DateToStr}(vs) \rrbracket_{\mathcal{S},\Gamma,\mathcal{T}} = \text{let } y = \text{z3.IntToStr}(vs[0]),$$
$$m = \text{ite}(vs[1] \leq 9, \text{``0''} + \text{z3.IntToStr}(vs[1]),$$
$$\text{z3.IntToStr}(vs[1])),$$
$$d = \text{ite}(vs[2] \leq 9, \text{``0''} + \text{z3.IntToStr}(vs[2]),$$
$$\text{z3.IntToStr}(vs[2]))$$
$$\text{in ite}(vs = \text{Null}, \text{Null}, y + \text{``-''} + m + \text{``-''} + d)$$

$$\llbracket \text{IntToDate}(v) \rrbracket_{\mathcal{S},\Gamma,\mathcal{T}} = \text{let } y = \text{fdiv}(v, 10^4), m = \text{fdiv}(v\%10^4, 10^2), d = v\%10^2,$$
$$\Phi_0 = y\%4 = 0 \wedge (y\%100 \neq 0 \vee y\%400 = 0)$$
$$\Phi_1 = \text{MIN\_YEAR} \leq y \leq \text{MAX\_YEAR},$$
$$\Phi_2 = 1 \leq m \leq 12,$$
$$\Phi_3 = 1 \leq d \wedge (\vee_{c \in \{1,3,5,7,8,10,12\}} m = c \to d \leq 31)$$
$$\wedge (m = 2 \to d \leq 28 + \text{ite}(\Phi_0, 1, 0))$$
$$\wedge (\vee_{c \in \{4,6,9,11\}} m = c \to d \leq 30)$$
$$\text{in ite}(v = \text{Null} \vee \neg(\Phi_1 \wedge \Phi_2 \wedge \Phi_3), \text{Null}, [y, m, d])$$

$$\llbracket \text{StrToDate}(s) \rrbracket_{\mathcal{S},\Gamma,\mathcal{T}} = \text{let } v = \llbracket \text{StrToInt}(s) \rrbracket_{\mathcal{S},\Gamma,\mathcal{T}} \text{ in}$$
$$\text{ite}(s = \text{Null}, \text{Null}, \llbracket \text{IntToDate}(v) \rrbracket_{\mathcal{S},\Gamma,\mathcal{T}})$$

$$\llbracket E_1 \diamond E_2 \rrbracket_{\mathcal{S},\Gamma,\mathcal{T}} = \text{let } v_1 = \llbracket \text{ToInt}(E_1) \rrbracket_{\mathcal{S},\Gamma,\mathcal{T}} \text{ and } v_2 = \llbracket \text{ToInt}(E_2) \rrbracket_{\mathcal{S},\Gamma,\mathcal{T}},$$
$$\text{in ite}(v_1 = \text{Null} \vee v_2 = \text{Null}, \text{Null}, v_1 \diamond v_2)$$

$$\llbracket \text{SubStr}(E_1, E_2, E_3) \rrbracket_{\mathcal{S},\Gamma,\mathcal{T}} = \text{let}$$
$$e_i = \llbracket E_i \rrbracket_{\mathcal{S},\Gamma,\mathcal{T}}, e_1' = \llbracket \text{ToStr}(e_1) \rrbracket_{\mathcal{S},\Gamma,\mathcal{T}}, l = \text{z3.Length}(e_1'),$$
$$e_2' = \llbracket \text{ToInt}(e_2) \rrbracket_{\mathcal{S},\Gamma,\mathcal{T}}, e_3' = \llbracket \text{ToInt}(e_3) \rrbracket_{\mathcal{S},\Gamma,\mathcal{T}},$$
$$v = \text{ite}(-l \leq e_2' < 0, e_2 + l, \text{ite}(0 < e_2' \leq l, e_2' - 1, l + 1)),$$
$$s = \text{ite}(v = 0 \vee v < -l \vee v > l \vee e_3' \leq 0, \varepsilon,$$
$$\text{ite}(e_3' \geq l - v, e_1'[v : l], e_1'[v : 2v + e_3']))$$
$$\text{in ite}(e_1 = \text{Null} \vee \text{IsStr}(e_2) \vee \text{IsStr}(e_3), \text{Null}, s)$$

$$\llbracket \text{Concate}(E_1, E_2) \rrbracket_{\mathcal{S},\Gamma,\mathcal{T}} = \text{let } v_1 = \llbracket \text{ToStr}(E_1) \rrbracket_{\mathcal{S},\Gamma,\mathcal{T}} \text{ and } v_2 = \llbracket \text{ToStr}(E_2) \rrbracket_{\mathcal{S},\Gamma,\mathcal{T}} \text{ in}$$
$$\text{ite}(v_1 = \text{Null} \vee v_2 = \text{Null}, \text{Null}, \text{z3.Concat}(v_1, v_2))$$

$$\llbracket \text{Strftime}(\kappa, E) \rrbracket_{\mathcal{S},\Gamma,\mathcal{T}} = \text{let } v = \llbracket \text{ToDate}(E) \rrbracket_{\mathcal{S},\Gamma,\mathcal{T}} \text{ in}$$
$$\text{ite}(v = \text{Null}, \text{Null},$$
$$\text{ite}(\kappa = \text{``\%Y''}, v[0], \text{ite}(\kappa = \text{``\%M''}, v[1], v[2])))$$

$$\llbracket \text{JulianDay}(E) \rrbracket_{\mathcal{S},\Gamma,\mathcal{T}} = \text{let } v = \llbracket E \rrbracket_{\mathcal{S},\Gamma,\mathcal{T}}, y = \text{ite}(v[1] \leq 2, v[0] - 1, v[0]),$$
$$m = \text{ite}(v[1] \leq 2, v[1] + 12, v[1]), d = v[2],$$
$$c = 2 - \text{fdiv}(y, 100) + \text{fdiv}(y, 400),$$
$$a_1 = \text{fdiv}(36525 * (y + 4716), 10^2),$$
$$a_2 = \text{fdiv}(306001 * (m + 1), 10^4),$$
$$\text{in } a_1 + a_2 + d + c - 1524.5, \text{ if IsDate}(v)$$

$$\llbracket \text{DateShift}(E, i, \delta) \rrbracket_{\mathcal{S},\Gamma,\mathcal{T}} = \text{let } v = \llbracket E \rrbracket_{\mathcal{S},\Gamma,\mathcal{T}} \text{ in}$$
$$\text{ite}(\delta = \text{``Year''}, \text{DateShiftByYears}(v, i),$$
$$\text{ite}(\delta = \text{``Month''}, \text{DateShiftByMonths}(v, i),$$
$$\text{DateShiftByDays}(v, i)))$$

Figure 18: Symbolic encoding for extended expressions. The floor division function is defined as $\text{fdiv}(x, y) = \text{ite}(x\%y = 0, x/y, (x - x\%y)/y)$. For clarity, we overload IsInt, IsStr and IsDate to check whether formulas represent integers, strings and dates, respectively. Type conversions and string manipulations are handled using the built-in functions of Z3.

$$
\begin{aligned}
[\![\mathsf{PrefixOf}(s, E)]\!]_{\mathcal{S},\Gamma,\mathcal{T}} &= \text{let } v = [\![\mathsf{ToStr}(E)]\!]_{\mathcal{S},\Gamma,\mathcal{T}} \text{ in ite}(v = \mathsf{Null}, \mathsf{Null}, \mathsf{z3.PrefixOf}(s, v)) \\
[\![\mathsf{SuffixOf}(s, E)]\!]_{\mathcal{S},\Gamma,\mathcal{T}} &= \text{let } v = [\![\mathsf{ToStr}(E)]\!]_{\mathcal{S},\Gamma,\mathcal{T}} \text{ in ite}(v = \mathsf{Null}, \mathsf{Null}, \mathsf{z3.SuffixOf}(s, v)) \\
[\![\mathsf{Like}(s, E)]\!]_{\mathcal{S},\Gamma,\mathcal{T}} &= \text{let } v = [\![\mathsf{ToStr}(E)]\!]_{\mathcal{S},\Gamma,\mathcal{T}} \text{ in} \\
&\quad \text{ite}(v = \mathsf{Null}, \mathsf{Null}, \mathsf{z3.RegexMatch}(s, v)) \\
[\![\mathsf{Contain}(s, E)]\!]_{\mathcal{S},\Gamma,\mathcal{T}} &= \text{let } v = [\![\mathsf{ToStr}(E)]\!]_{\mathcal{S},\Gamma,\mathcal{T}} \text{ and } s' = \mathsf{Concat}(\text{``.*''}, s, \text{``.*''}) \text{ in} \\
&\quad \text{ite}(v = \mathsf{Null}, \mathsf{Null}, \mathsf{z3.RegexMatch}(s', v)) \\
[\![E_1 \odot E_2]\!]_{\mathcal{S},\Gamma,\mathcal{T}} &= \text{let } v_1 = [\![E_1]\!]_{\mathcal{S},\Gamma,\mathcal{T}} \text{ and } v_2 = [\![E_2]\!]_{\mathcal{S},\Gamma,\mathcal{T}} \text{ in} \\
&\quad \text{ite}(v_1 = \mathsf{Null} \vee v_2 = \mathsf{Null}, \bot, v_1 \odot v_2), \text{ if } \mathsf{Type}(v_1) = \mathsf{Type}(v_2)
\end{aligned}
$$

Figure 19: Symbolic encoding for extended predicates.

## G  PROOF

In this section, we provide the proof of theorems in the main paper.

**Theorem 1** (Correctness of expression encoding). *Let $D$ be a database over schema $\mathcal{S}$, $xs$ be a tuple list, and $E$ be an expression. Consider a symbolic database $\Gamma$ over $\mathcal{S}$, a list of symbolic tuples $\mathcal{T}$, and $E$'s symbolic encoding $[\![E]\!]_{\mathcal{S},\Gamma,\mathcal{T}}$. For any satisfying interpretation $\mathcal{I}$ with $\mathcal{I}(\Gamma) = D \wedge \mathcal{I}(\mathcal{T}) = xs$, evaluating the expression $E$ over the database $D$ and the tuple list $xs$ yields the interpretation of $E$'s symbolic encoding $\mathcal{I}([\![E]\!]_{\mathcal{S},\Gamma,\mathcal{T}})$, i.e., $\mathcal{I}(\Gamma) = D \wedge \mathcal{I}(\mathcal{T}) = xs \Rightarrow [\![E]\!]_{D,xs} = \mathcal{I}([\![E]\!]_{\mathcal{S},\Gamma,\mathcal{T}})$.*

**Lemma 1.** *Suppose $[\![E]\!]_{D,xs} = v$, then $\mathcal{I}(\Gamma) = D \wedge \mathcal{I}(\mathcal{T}) = xs \Rightarrow [\![E]\!]_{\mathcal{I}(\Gamma),\mathcal{I}(\mathcal{T})} = \mathcal{I}([\![E]\!]_{\mathcal{S},\Gamma,\mathcal{T}})$ is true iff $[\![E]\!]_{\mathcal{I}(\Gamma),\mathcal{I}(\mathcal{T})} = v$ and $\mathcal{I}([\![E]\!]_{\mathcal{S},\Gamma,\mathcal{T}}) = v$.*

*Proof.* Theorem 1 is proved by proving Lemma 1. By structural induction on $E$.

1. Base cases and some inductive cases are proved in He et al. (2024).

2. Inductive case: $E = \mathsf{ToInt}(E)$

   $[\![\mathsf{ToInt}(E)]\!]_{\mathcal{S},\Gamma,\mathcal{T}} = \text{ite}(v = \mathsf{Null} \vee \mathsf{IsInt}(v), v, \text{ite}(\mathsf{IsStr}(v), [\![\mathsf{StrToInt}(v)]\!]_{\mathcal{S},\Gamma,\mathcal{T}},$
   $[\![\mathsf{DateToInt}(v)]\!]_{\mathcal{S},\Gamma,\mathcal{T}}))$ where $v = [\![E]\!]_{\mathcal{S},\Gamma,\mathcal{T}}$ by Figure 18. $[\![\mathsf{ToInt}(E)]\!]_{\mathcal{I}(\Gamma),\mathcal{I}(\mathcal{T})} = \text{ite}(v' = \mathsf{Null} \vee$
   $\mathsf{IsInt}(v'), v', \text{ite}(\mathsf{IsStr}(v'), [\![\mathsf{StrToInt}(v')]\!]_{\mathcal{I}(\Gamma),\mathcal{I}(\mathcal{T})}, [\![\mathsf{DateToInt}(v')]\!]_{\mathcal{I}(\Gamma),\mathcal{I}(\mathcal{T})}))$
   where $v' = [\![E]\!]_{\mathcal{I}(\Gamma),\mathcal{I}(\mathcal{T})}$ by Figure 17. By inductive hypothesis, we have
   $\mathcal{I}(v) = \mathcal{I}([\![E]\!]_{\mathcal{S},\Gamma,\mathcal{T}}) = [\![E]\!]_{\mathcal{I}(\Gamma),\mathcal{I}(\mathcal{T})} = v'$. Therefore,

   $$
   \begin{aligned}
   \mathcal{I}([\![\mathsf{ToInt}(E)]\!]_{\mathcal{S},\Gamma,\mathcal{T}}) &= \mathcal{I}(\text{ite}(v = \mathsf{Null} \vee \mathsf{IsInt}(v), v, \text{ite}(\mathsf{IsStr}(v), [\![\mathsf{StrToInt}(v)]\!]_{\mathcal{S},\Gamma,\mathcal{T}}, \\
   &\qquad [\![\mathsf{DateToInt}(v)]\!]_{\mathcal{S},\Gamma,\mathcal{T}}))) \\
   &= \text{ite}(\mathcal{I}(v) = \mathsf{Null} \vee \mathcal{I}(\mathsf{IsInt}(v)), \mathcal{I}(v), \text{ite}(\mathcal{I}(\mathsf{IsStr}(v)), \\
   &\qquad \mathcal{I}([\![\mathsf{StrToInt}(v)]\!]_{\mathcal{S},\Gamma,\mathcal{T}}), \mathcal{I}([\![\mathsf{DateToInt}(v)]\!]_{\mathcal{S},\Gamma,\mathcal{T}}))) \\
   &= \text{ite}(\mathcal{I}(v) = \mathsf{Null} \vee \mathsf{IsInt}(\mathcal{I}(v)), \mathcal{I}(v), \text{ite}(\mathsf{IsStr}(\mathcal{I}(v)), \\
   &\qquad [\![\mathsf{StrToInt}(\mathcal{I}(v))]\!]_{\mathcal{I}(\Gamma),\mathcal{I}(\mathcal{T})}, [\![\mathsf{DateToInt}(\mathcal{I}(v))]\!]_{\mathcal{I}(\Gamma),\mathcal{I}(\mathcal{T})})) \\
   &= \text{ite}(v' = \mathsf{Null} \vee \mathsf{IsInt}(v'), v', \text{ite}(\mathsf{IsStr}(v'), \\
   &\qquad [\![\mathsf{StrToInt}(v')]\!]_{\mathcal{I}(\Gamma),\mathcal{I}(\mathcal{T})}, [\![\mathsf{DateToInt}(v')]\!]_{\mathcal{I}(\Gamma),\mathcal{I}(\mathcal{T})})) \\
   &= [\![\mathsf{ToInt}(E)]\!]_{\mathcal{I}(\Gamma),\mathcal{I}(\mathcal{T})}
   \end{aligned}
   $$

3. Inductive case: $E = \mathsf{ToDate}(E)$

   $[\![\mathsf{ToDate}(E)]\!]_{\mathcal{S},\Gamma,\mathcal{T}} = \text{ite}(v = \mathsf{Null} \vee \mathsf{IsDate}(v), v, \text{ite}(\mathsf{IsInt}(v), [\![\mathsf{IntToDate}(v)]\!]_{\mathcal{S},\Gamma,\mathcal{T}},$
   $[\![\mathsf{StrToDate}(v)]\!]_{\mathcal{S},\Gamma,\mathcal{T}}))$ where $v = [\![E]\!]_{\mathcal{S},\Gamma,\mathcal{T}}$ by Figure 18. $[\![\mathsf{ToDate}(E)]\!]_{\mathcal{I}(\Gamma),\mathcal{I}(\mathcal{T})} = \text{ite}(v' = \mathsf{Null} \vee$
   $\mathsf{IsDate}(v'), v', \text{ite}(\mathsf{IsInt}(v'), [\![\mathsf{IntToDate}(v')]\!]_{\mathcal{I}(\Gamma),\mathcal{I}(\mathcal{T})}, [\![\mathsf{StrToDate}(v')]\!]_{\mathcal{I}(\Gamma),\mathcal{I}(\mathcal{T})}))$
   where $v' = [\![E]\!]_{\mathcal{I}(\Gamma),\mathcal{I}(\mathcal{T})}$ by Figure 17. By inductive hypothesis, we have

$\mathcal{I}(v) = \mathcal{I}(\llbracket E \rrbracket_{\mathcal{S},\Gamma,\mathcal{T}}) = \llbracket E \rrbracket_{\mathcal{I}(\Gamma),\mathcal{I}(\mathcal{T})} = v'$. Therefore,

$$
\begin{aligned}
\mathcal{I}(\llbracket \mathsf{ToDate}(E) \rrbracket_{\mathcal{S},\Gamma,\mathcal{T}}) &= \mathcal{I}(\mathsf{ite}(v = \mathsf{Null} \vee \mathsf{IsDate}(v), v, \mathsf{ite}(\mathsf{IsInt}(v), \\
&\qquad \llbracket \mathsf{IntToDate}(v) \rrbracket_{\mathcal{S},\Gamma,\mathcal{T}}, \llbracket \mathsf{StrToDate}(v) \rrbracket_{\mathcal{S},\Gamma,\mathcal{T}}))) \\
&= \mathsf{ite}(\mathcal{I}(v) = \mathsf{Null} \vee \mathcal{I}(\mathsf{IsDate}(v)), \mathcal{I}(v), \mathsf{ite}(\mathcal{I}(\mathsf{IsInt}(v)), \\
&\qquad \mathcal{I}(\llbracket \mathsf{IntToDate}(v) \rrbracket_{\mathcal{S},\Gamma,\mathcal{T}}), \mathcal{I}(\llbracket \mathsf{StrToDate}(v) \rrbracket_{\mathcal{S},\Gamma,\mathcal{T}}))) \\
&= \mathsf{ite}(\mathcal{I}(v) = \mathsf{Null} \vee \mathsf{IsDate}(\mathcal{I}(v)), \mathcal{I}(v), \mathsf{ite}(\mathsf{IsInt}(\mathcal{I}(v)), \\
&\qquad \mathcal{I}(\llbracket \mathsf{IntToDate}(v) \rrbracket_{\mathcal{S},\Gamma,\mathcal{T}}), \mathcal{I}(\llbracket \mathsf{StrToDate}(v) \rrbracket_{\mathcal{S},\Gamma,\mathcal{T}}))) \\
&= \mathsf{ite}(v' = \mathsf{Null} \vee \mathsf{IsDate}(v'), v', \mathsf{ite}(\mathsf{IsInt}(v'), \\
&\qquad \llbracket \mathsf{IntToDate}(v') \rrbracket_{\mathcal{I}(\Gamma),\mathcal{I}(\mathcal{T})}, \llbracket \mathsf{StrToDate}(v') \rrbracket_{\mathcal{I}(\Gamma),\mathcal{I}(\mathcal{T})})) \\
&= \llbracket \mathsf{ToDate}(E) \rrbracket_{\mathcal{I}(\Gamma),\mathcal{I}(\mathcal{T})}
\end{aligned}
$$

4. Inductive case: $E = \mathsf{ToStr}(E)$

$\llbracket \mathsf{ToStr}(E) \rrbracket_{\mathcal{S},\Gamma,\mathcal{T}} = \mathsf{ite}(v = \mathsf{Null} \vee \mathsf{IsStr}(v), v, \mathsf{ite}(\mathsf{IsInt}(v), \llbracket \mathsf{IntToStr}(v) \rrbracket_{\mathcal{S},\Gamma,\mathcal{T}}, \llbracket \mathsf{DateToStr}(v) \rrbracket_{\mathcal{S},\Gamma,\mathcal{T}}))$ where $v = \llbracket E \rrbracket_{\mathcal{S},\Gamma,\mathcal{T}}$ by Figure 18. $\llbracket \mathsf{ToStr}(E) \rrbracket_{\mathcal{I}(\Gamma),\mathcal{I}(\mathcal{T})} = \mathsf{ite}(v' = \mathsf{Null} \vee \mathsf{IsStr}(v'), v', \mathsf{ite}(\mathsf{IsInt}(v'), \llbracket \mathsf{IntToStr}(v') \rrbracket_{\mathcal{I}(\Gamma),\mathcal{I}(\mathcal{T})}, \llbracket \mathsf{DateToStr}(v') \rrbracket_{\mathcal{I}(\Gamma),\mathcal{I}(\mathcal{T})}))$ where $v' = \llbracket E \rrbracket_{\mathcal{I}(\Gamma),\mathcal{I}(\mathcal{T})}$ by Figure 17. By inductive hypothesis, we have $\mathcal{I}(v) = \mathcal{I}(\llbracket E \rrbracket_{\mathcal{S},\Gamma,\mathcal{T}}) = \llbracket E \rrbracket_{\mathcal{I}(\Gamma),\mathcal{I}(\mathcal{T})} = v'$. Therefore,

$$
\begin{aligned}
\mathcal{I}(\llbracket \mathsf{ToStr}(E) \rrbracket_{\mathcal{S},\Gamma,\mathcal{T}}) &= \mathcal{I}(\mathsf{ite}(v = \mathsf{Null} \vee \mathsf{IsStr}(v), v, \mathsf{ite}(\mathsf{IsInt}(v), \\
&\qquad \llbracket \mathsf{IntToStr}(v) \rrbracket_{\mathcal{S},\Gamma,\mathcal{T}}, \llbracket \mathsf{DateToStr}(v) \rrbracket_{\mathcal{S},\Gamma,\mathcal{T}}))) \\
&= \mathsf{ite}(\mathcal{I}(v) = \mathsf{Null} \vee \mathcal{I}(\mathsf{IsStr}(v)), \mathcal{I}(v), \mathsf{ite}(\mathcal{I}(\mathsf{IsInt}(v)), \\
&\qquad \mathcal{I}(\llbracket \mathsf{IntToStr}(v) \rrbracket_{\mathcal{S},\Gamma,\mathcal{T}}), \mathcal{I}(\llbracket \mathsf{DateToStr}(v) \rrbracket_{\mathcal{S},\Gamma,\mathcal{T}}))) \\
&= \mathsf{ite}(\mathcal{I}(v) = \mathsf{Null} \vee \mathsf{IsStr}(\mathcal{I}(v)), \mathcal{I}(v), \mathsf{ite}(\mathsf{IsInt}(\mathcal{I}(v)), \\
&\qquad \mathcal{I}(\llbracket \mathsf{IntToStr}(v) \rrbracket_{\mathcal{S},\Gamma,\mathcal{T}}), \mathcal{I}(\llbracket \mathsf{DateToStr}(v) \rrbracket_{\mathcal{S},\Gamma,\mathcal{T}}))) \\
&= \mathsf{ite}(v' = \mathsf{Null} \vee \mathsf{IsStr}(v'), v', \mathsf{ite}(\mathsf{IsInt}(v'), \\
&\qquad \llbracket \mathsf{IntToStr}(v') \rrbracket_{\mathcal{I}(\Gamma),\mathcal{I}(\mathcal{T})}, \llbracket \mathsf{DateToStr}(v') \rrbracket_{\mathcal{I}(\Gamma),\mathcal{I}(\mathcal{T})})) \\
&= \llbracket \mathsf{ToStr}(E) \rrbracket_{\mathcal{I}(\Gamma),\mathcal{I}(\mathcal{T})}
\end{aligned}
$$

5. Inductive case: $E = \mathsf{DateToInt}(vs)$

$\llbracket \mathsf{DateToInt}(vs) \rrbracket_{\mathcal{S},\Gamma,\mathcal{T}} = \mathsf{ite}(vs = \mathsf{Null}, \mathsf{Null}, vs[0] * 10^4 + vs[1] * 10^2 + vs[2])$ by Figure 18. $\llbracket \mathsf{DateToInt}(vs) \rrbracket_{\mathcal{I}(\Gamma),\mathcal{I}(\mathcal{T})} = \mathsf{ite}(vs = \mathsf{Null}, \mathsf{Null}, vs[0] * 10^4 + vs[1] * 10^2 + vs[2])$ by Figure 17. Therefore, $\mathcal{I}(\llbracket \mathsf{DateToInt}(vs) \rrbracket_{\mathcal{S},\Gamma,\mathcal{T}}) = \mathsf{ite}(vs = \mathsf{Null}, \mathsf{Null}, vs[0] * 10^4 + vs[1] * 10^2 + vs[2]) = \llbracket \mathsf{DateToInt}(vs) \rrbracket_{\mathcal{I}(\Gamma),\mathcal{I}(\mathcal{T})}$.

6. Inductive case: $E = \mathsf{StrToInt}(s)$

$\llbracket \mathsf{StrToInt}(s) \rrbracket_{\mathcal{S},\Gamma,\mathcal{T}} = \mathsf{ite}(s = \mathsf{Null}, \mathsf{Null}, \mathsf{ite}(\Phi, v, 0))$ where $s_1 = s[1 : \mathrm{z3.Length}(s)]$, $v_1 = \mathrm{z3.StrToInt}(s_1)$, $v = \mathsf{ite}(s[0] = \text{"-"}, -v_1, \mathrm{z3.StrToInt}(s))$, and $\Phi = \mathsf{ite}(v < 0, \mathrm{z3.IntToStr}(-v) = v_1, \mathrm{z3.IntToStr}(v) = s)$ by Figure 18. $\llbracket \mathsf{StrToInt}(s) \rrbracket_{\mathcal{I}(\Gamma),\mathcal{I}(\mathcal{T})} = \mathsf{ite}(v' = \mathsf{Null}, \mathsf{Null}, v')$ where $v' = \mathsf{ite}(\mathsf{IsDigits}(s), \mathrm{StrToInt}(s), \mathsf{ite}(s[0] = \text{"-"} \wedge \mathsf{IsDigits}(s[1:]), -\mathrm{StrToInt}(s), 0))$ by Figure 17.

On the one hand, the Z3 builtin function $\mathrm{z3.StrToInt}(s) = \mathrm{StrToInt}(s)$ if $\mathrm{StrToInt}(s) \geq 0$; otherwise, $\mathrm{z3.StrToInt}(s) = -1$. To show our encoding precisely capture semantics of *SQL's type conversion from strings to integers*, let us discuss it in three cases:

(a) If $\mathrm{StrToInt}(s) \geq 0$, then $v = \mathrm{z3.StrToInt}(s) = \mathrm{StrToInt}(s)$ and $\Phi$ holds. Thus, $\mathsf{ite}(\Phi, v, 0) = v$.

(b) If $\mathrm{StrToInt}(s) < 0$, then $v = -v_1$ and $\Phi = \top$ where $v_1 = \mathrm{StrToInt}(s[1:])$. $\mathsf{ite}(\Phi, v, 0) = v = -v_1$.

(c) If $s$ contains more than digits (e.g., "abc" and "-abc"), MYSQL evaluates non-numerical strings to 0 by default. By the semantics of z3.StrToInt, $\Phi$ never holds which leads $\mathsf{ite}(\Phi, v, 0) = 0$.

By 6a, 6c and 6c, we known $\mathsf{ite}(\Phi, s, 0)$ precisely captures the semantics of *SQL's type conversion from strings to integers*.

On the other hand, let us discuss the rule in three cases:

(a) If $\text{StrToInt}(s) \geq 0$, then $v' = \text{StrToInt}(s)$.

(b) If $\text{StrToInt}(s) < 0$, then $v' = -\text{StrToInt}(s[1\,:])$.

(c) If $s$ contains more than digits (e.g., "abc" and "-abc"), MYSQL evaluates non-numerical strings to 0 by default. By the semantics of this rule, $v' = 0$.

By 6a, 6c and 6c, we known $v'$ precisely captures the semantics of *SQL's type conversion from strings to integers*.

Therefore, $\mathcal{I}(\text{ite}(\Phi, s, 0)) = v'$ and

$$
\begin{aligned}
\mathcal{I}([\![\text{StrToInt}(s)]\!]_{\mathcal{S},\Gamma,\mathcal{T}}) &= \mathcal{I}(\text{ite}(s = \text{Null}, \text{Null}, \text{ite}(\Phi, v, 0))) \\
&= \text{ite}(s = \text{Null}, \text{Null}, \mathcal{I}(\text{ite}(\Phi, v, 0))) \\
&= \text{ite}(s = \text{Null}, \text{Null}, v') \\
&= [\![\text{StrToInt}(s)]\!]_{\mathcal{I}(\Gamma),\mathcal{I}(\mathcal{T})}
\end{aligned}
$$

7. Inductive case: $E = \text{IntToStr}(v)$

$[\![\text{IntToStr}(v)]\!]_{\mathcal{S},\Gamma,\mathcal{T}} = \text{ite}(v = \text{Null}, \text{Null}, \text{z3.IntToStr}(v))$ by Figure 18. $[\![\text{IntToStr}(v)]\!]_{\mathcal{I}(\Gamma),\mathcal{I}(\mathcal{T})} = \text{ite}(v = \text{Null}, \text{Null}, \text{IntToStr}(v))$ by Figure 17. Note that since the Z3 builtin function z3.IntToStr precisely capture the semantics of IntToStr, $\mathcal{I}(\text{z3.IntToStr}(v)) = \text{IntToStr}(v)$. Therefore,

$$
\begin{aligned}
\mathcal{I}([\![\text{IntToStr}(v)]\!]_{\mathcal{S},\Gamma,\mathcal{T}}) &= \mathcal{I}(\text{ite}(v = \text{Null}, \text{Null}, \text{z3.IntToStr}(v))) \\
&= \text{ite}(v = \text{Null}, \text{Null}, \mathcal{I}(\text{z3.IntToStr}(v))) \\
&= \text{ite}(v = \text{Null}, \text{Null}, \text{IntToStr}(v)) \\
&= [\![\text{IntToStr}(v)]\!]_{\mathcal{I}(\Gamma),\mathcal{I}(\mathcal{T})}
\end{aligned}
$$

8. Inductive case: $E = \text{DateToStr}(vs)$

$[\![\text{DateToStr}(vs)]\!]_{\mathcal{S},\Gamma,\mathcal{T}} = \text{ite}(vs = \text{Null}, \text{Null}, y + \text{"-"} + m + \text{"-"} + d)$ where $y = \text{z3.IntToStr}(vs[0])$, $m = \text{ite}(vs[1] \leq 9, \text{"0"} + \text{z3.IntToStr}(vs[1]), \text{z3.IntToStr}(vs[1]))$, and $d = \text{ite}(vs[2] \leq 9, \text{"0"} + \text{z3.IntToStr}(vs[2]), \text{z3.IntToStr}(vs[2]))$ by Figure 18. $[\![\text{DateToStr}(vs)]\!]_{\mathcal{I}(\Gamma),\mathcal{I}(\mathcal{T})} = \text{ite}(vs = \text{Null}, \text{Null}, y' + \text{"-"} + m' + \text{"-"} + d')$ where $y' = \text{IntToStr}(vs[0])$, $m' = \text{ite}(vs[1] \leq 9, \text{"0"} + \text{IntToStr}(vs[1]), \text{IntToStr}(vs[1]))$, and $d' = \text{ite}(vs[2] \leq 9, \text{"0"} + \text{IntToStr}(vs[2]), \text{IntToStr}(vs[2]))$ by Figure 17. Note that since the Z3 builtin function z3.IntToStr precisely capture the semantics of IntToStr, $\mathcal{I}(y) = y'$, $\mathcal{I}(m) = m'$, and $\mathcal{I}(d) = d'$. Therefore,

$$
\begin{aligned}
\mathcal{I}([\![\text{DateToStr}(vs)]\!]_{\mathcal{S},\Gamma,\mathcal{T}}) &= \mathcal{I}(\text{ite}(vs = \text{Null}, \text{Null}, y + \text{"-"} + m + \text{"-"} + d)) \\
&= \text{ite}(vs = \text{Null}, \text{Null}, \mathcal{I}(y) + \text{"-"} + \mathcal{I}(m) + \text{"-"} + \mathcal{I}(d)) \\
&= \text{ite}(vs = \text{Null}, \text{Null}, y' + \text{"-"} + m' + \text{"-"} + d') \\
&= [\![\text{DateToStr}(v)]\!]_{\mathcal{I}(\Gamma),\mathcal{I}(\mathcal{T})}
\end{aligned}
$$

9. Inductive case: $E = \text{IntToDate}(v)$

$[\![\text{IntToDate}(v)]\!]_{\mathcal{S},\Gamma,\mathcal{T}} = \text{ite}(v = \text{Null} \vee \neg(\Phi_1 \wedge \Phi_2 \wedge \Phi_3), \text{Null}, [y, m, d])$ where $\text{fdiv}(x, y) = \text{ite}(x\%y = 0, x/y, (x - x\%y)/y)$, $y = \text{fdiv}(v, 10^4)$, $m = \text{fdiv}(v\%10^4, 10^2)$, $d = v\%10^2$, $\Phi_0 = y\%4 = 0 \wedge (y\%100 \neq 0 \vee y\%400 = 0)$, $\Phi_1 = \text{MIN\_YEAR} \leq y \leq \text{MAX\_YEAR}$, $\Phi_2 = 1 \leq m \leq 12$, $\Phi_3 = 1 \leq d \wedge (\vee_{c \in \{1,3,5,7,8,10,12\}} m = c \rightarrow d \leq 31) \wedge (m = 2 \rightarrow d \leq 28 + \text{ite}(\Phi_0, 1, 0)) \wedge (\vee_{c \in \{4,6,9,11\}} m = c \rightarrow d \leq 30)$ by Figure 18. $[\![\text{IntToDate}(v)]\!]_{\mathcal{I}(\Gamma),\mathcal{I}(\mathcal{T})} = \text{ite}(v' = \text{Null} \vee \text{IsValidDate}(v), \text{Null}, [v_1', v_2', v_3'])$ where $v_1' = \lfloor v/10^4 \rfloor$, $v_2' = \lfloor (v\%10^4)/10^2 \rfloor$, $v_3' = v\%10^2$ by Figure 17. By semantics of fdiv, we know $y = v_1'$, $m = v_2'$ and $d = v_3'$. Note that the function IsValidDate precisely capture the semantics of $\neg(\Phi_1 \wedge \Phi_2 \wedge \Phi_3)$, checking whether a date is valid in MYSQL. Therefore, $\mathcal{I}(\neg(\Phi_1 \wedge \Phi_2 \wedge \Phi_3)) = \text{IsValidDate}(v')$ and

$$
\begin{aligned}
\mathcal{I}([\![\text{IntToDate}(v)]\!]_{\mathcal{S},\Gamma,\mathcal{T}}) &= \mathcal{I}(\text{ite}(v = \text{Null} \vee \neg(\Phi_1 \wedge \Phi_2 \wedge \Phi_3), \text{Null}, [y, m, d])) \\
&= \text{ite}(v = \text{Null} \vee \mathcal{I}(\neg(\Phi_1 \wedge \Phi_2 \wedge \Phi_3)), \text{Null}, \mathcal{I}([y, m, d])) \\
&= \text{ite}(v = \text{Null} \vee \text{IsValidDate}(v'), \text{Null}, [v_1', v_2', v_3']) \\
&= [\![\text{IntToDate}(v)]\!]_{\mathcal{I}(\Gamma),\mathcal{I}(\mathcal{T})}
\end{aligned}
$$

10. Inductive case: $E = \mathsf{StrToDate}(s)$

$[\![\mathsf{StrToDate}(s)]\!]_{\mathcal{S},\Gamma,\mathcal{T}} = \mathsf{ite}(s = \mathrm{Null}, \mathrm{Null}, [\![\mathsf{IntToDate}(v)]\!]_{\mathcal{S},\Gamma,\mathcal{T}})$ where $v = [\![\mathsf{StrToInt}(s)]\!]_{\mathcal{S},\Gamma,\mathcal{T}}$ by Figure 18. $[\![\mathsf{StrToDate}(s)]\!]_{\mathcal{I}(\Gamma),\mathcal{I}(\mathcal{T})} = \mathsf{ite}(s = \mathrm{Null}, \mathrm{Null}, [\![\mathsf{IntToDate}(v')]\!]_{\mathcal{I}(\Gamma),\mathcal{I}(\mathcal{T})})$ where $v' = [\![\mathsf{StrToInt}(s)]\!]_{\mathcal{S},\Gamma,\mathcal{T}}$ by Figure 17. By inductive hypothesis, we have $\mathcal{I}([\![\mathsf{IntToDate}(v)]\!]_{\mathcal{S},\Gamma,\mathcal{T}}) = [\![\mathcal{I}(\mathsf{IntToDate}(v))]\!]_{\mathcal{I}(\Gamma),\mathcal{I}(\mathcal{T})} = [\![\mathsf{IntToDate}(\mathcal{I}(v))]\!]_{\mathcal{I}(\Gamma),\mathcal{I}(\mathcal{T})} = [\![\mathsf{IntToDate}(v')]\!]_{\mathcal{I}(\Gamma),\mathcal{I}(\mathcal{T})}$. Therefore,

$$
\begin{aligned}
\mathcal{I}([\![\mathsf{StrToDate}(s)]\!]_{\mathcal{S},\Gamma,\mathcal{T}}) &= \mathcal{I}(\mathsf{ite}(s = \mathrm{Null}, \mathrm{Null}, [\![\mathsf{IntToDate}(v)]\!]_{\mathcal{S},\Gamma,\mathcal{T}})) \\
&= \mathsf{ite}(s = \mathrm{Null}, \mathrm{Null}, \mathcal{I}([\![\mathsf{IntToDate}(v)]\!]_{\mathcal{S},\Gamma,\mathcal{T}})) \\
&= \mathsf{ite}(s = \mathrm{Null}, \mathrm{Null}, [\![\mathsf{IntToDate}(v')]\!]_{\mathcal{I}(\Gamma),\mathcal{I}(\mathcal{T})}) \\
&= [\![\mathsf{StrToDate}(s)]\!]_{\mathcal{I}(\Gamma),\mathcal{I}(\mathcal{T})}
\end{aligned}
$$

11. Inductive case: $E = E_1 \diamond E_2$.

Since our extended grammar considers Null, integers, dates and strings, as shown in Figure 2, the proof for this inductive case is overloaded.

$[\![E_1 \diamond E_2]\!]_{\mathcal{S},\Gamma,\mathcal{T}} = \mathsf{ite}(v_1 = \mathrm{Null} \vee v_2 = \mathrm{Null}, \mathrm{Null}, v_1 \diamond v_2)$ where $v_1 = [\![\mathsf{ToInt}(E_1)]\!]_{\mathcal{S},\Gamma,\mathcal{T}}$ and $v_2 = [\![\mathsf{ToInt}(E_2)]\!]_{\mathcal{S},\Gamma,\mathcal{T}}$ by Figure 18. $[\![E_1 \diamond E_2]\!]_{\mathcal{I}(\Gamma),\mathcal{I}(\mathcal{T})} = \mathsf{ite}(v'_1 = \mathrm{Null} \vee v'_2 = \mathrm{Null}, \mathrm{Null}, v'_1 \diamond v'_2)$ where $v'_1 = [\![\mathsf{ToInt}(E_1)]\!]_{\mathcal{I}(\Gamma),\mathcal{I}(\mathcal{T})}$ and $v'_2 = [\![\mathsf{ToInt}(E_2)]\!]_{\mathcal{I}(\Gamma),\mathcal{I}(\mathcal{T})}$ by Figure 17. By inductive hypothesis, we have $\mathcal{I}(v_1) = \mathcal{I}([\![\mathsf{ToInt}(E_1)]\!]_{\mathcal{S},\Gamma,\mathcal{T}}) = [\![\mathsf{ToInt}(E_1)]\!]_{\mathcal{I}(\Gamma),\mathcal{I}(\mathcal{T})} = v'_1$ and $\mathcal{I}(v_2) = \mathcal{I}([\![\mathsf{ToInt}(E_2)]\!]_{\mathcal{S},\Gamma,\mathcal{T}}) = [\![\mathsf{ToInt}(E_2)]\!]_{\mathcal{I}(\Gamma),\mathcal{I}(\mathcal{T})} = v'_2$. Therefore,

$$
\begin{aligned}
\mathcal{I}([\![E_1 \diamond E_2]\!]_{\mathcal{S},\Gamma,\mathcal{T}}) &= \mathcal{I}(\mathsf{ite}(v_1 = \mathrm{Null} \vee v_2 = \mathrm{Null}, \mathrm{Null}, v_1 \diamond v_2)) \\
&= \mathsf{ite}(\mathcal{I}((v_1) = \mathrm{Null} \vee \mathcal{I}((v_2) = \mathrm{Null}, \mathrm{Null}, \mathcal{I}((v_1) \diamond \mathcal{I}((v_2))) \\
&= \mathsf{ite}(v'_1 = \mathrm{Null} \vee v'_2 = \mathrm{Null}, \mathrm{Null}, v'_1 \diamond v'_2) \\
&= [\![E_1 \diamond E_2]\!]_{\mathcal{I}(\Gamma),\mathcal{I}(\mathcal{T})}
\end{aligned}
$$

12. Inductive case: $E = \mathsf{SubStr}(E_1, E_2, E_3)$.

$[\![\mathsf{SubStr}(E_1, E_2, E_3)]\!]_{\mathcal{S},\Gamma,\mathcal{T}} = \mathsf{ite}(e_1 = \mathrm{Null} \vee \mathsf{IsStr}(e_2) \vee \mathsf{IsStr}(e_3), \mathrm{Null}, s)$ where $e_i = [\![E_i]\!]_{\mathcal{S},\Gamma,\mathcal{T}}$ for $1 \leq i \leq 3$, $e'_1 = [\![\mathsf{ToStr}(e_1)]\!]_{\mathcal{S},\Gamma,\mathcal{T}}$, $l = \mathsf{z3.Length}(e'_1)$, $e'_2 = [\![\mathsf{ToInt}(e_2)]\!]_{\mathcal{S},\Gamma,\mathcal{T}}$, $e'_3 = [\![\mathsf{ToInt}(e_3)]\!]_{\mathcal{S},\Gamma,\mathcal{T}}$, $v = \mathsf{ite}(-l \leq e_2 < 0, \mathsf{ite}(0 < e'_2 \leq l, e'_2 - 1, l + 1))$, $s = \mathsf{ite}(v = 0 \vee v < -l \vee v > l \vee e'_3 \leq 0, \varepsilon, \mathsf{ite}(e'_3 \geq l - v, e'_1[v : l], e'_1[v : 2v + e'_3]))$ by Figure 18.
$[\![\mathsf{SubStr}(E_1, E_2, E_3)]\!]_{\mathcal{I}(\Gamma),\mathcal{I}(\mathcal{T})} = \mathsf{ite}(e_4 = \mathrm{Null} \vee \mathsf{IsStr}(e_5) \vee \mathsf{IsStr}(e_6), \mathrm{Null}, s)$ where $e_{i+3} = [\![E_i]\!]_{\mathcal{I}(\Gamma),\mathcal{I}(\mathcal{T})}$ for $1 \leq i \leq 3$, $e'_4 = [\![\mathsf{ToStr}(e_4)]\!]_{\mathcal{I}(\Gamma),\mathcal{I}(\mathcal{T})}$, $l' = \mathsf{z3.Length}(e'_4)$, $e'_5 = [\![\mathsf{ToInt}(e_5)]\!]_{\mathcal{I}(\Gamma),\mathcal{I}(\mathcal{T})}$, $e'_6 = [\![\mathsf{ToInt}(e_6)]\!]_{\mathcal{I}(\Gamma),\mathcal{I}(\mathcal{T})}$, $v' = \mathsf{ite}(-l \leq e_5 < 0, \mathsf{ite}(0 < e'_5 \leq l, e'_5 - 1, l + 1))$, $s' = \mathsf{ite}(v = 0 \vee v < -l \vee v > l \vee e'_6 \leq 0, \varepsilon, \mathsf{ite}(e'_6 \geq l - v, e'_4[v : l], e'_1[v : 2v + e'_6]))$ by Figure 17.
By inductive hypothesis, we have $\mathcal{I}(e_i) = \mathcal{I}([\![E_i]\!]_{\mathcal{I}(\Gamma),\mathcal{I}(\mathcal{T})}) = [\![E_i]\!]_{\mathcal{S},\Gamma,\mathcal{T}} = e_{i+3}$ for $1 \leq i \leq 3$. Then, $\mathcal{I}(e'_1) = \mathcal{I}([\![\mathsf{ToStr}(e_1)]\!]_{\mathcal{S},\Gamma,\mathcal{T}}) = [\![\mathsf{ToStr}(e_4)]\!]_{\mathcal{I}(\Gamma),\mathcal{I}(\mathcal{T})} = e'_4$, $\mathcal{I}(e'_2) = \mathcal{I}([\![\mathsf{ToInt}(e_2)]\!]_{\mathcal{S},\Gamma,\mathcal{T}}) = [\![\mathsf{ToInt}(e_5)]\!]_{\mathcal{I}(\Gamma),\mathcal{I}(\mathcal{T})} = e'_5$, and $\mathcal{I}(e'_3) = \mathcal{I}([\![\mathsf{ToInt}(e_3)]\!]_{\mathcal{S},\Gamma,\mathcal{T}}) = [\![\mathsf{ToInt}(e_6)]\!]_{\mathcal{I}(\Gamma),\mathcal{I}(\mathcal{T})} = e'_6$, $\mathcal{I}(v) = v'$, and $\mathcal{I}(s) = s'$. Furthermore, since the Z3 builtin function z3.Length precisely captures the semantics of len, we have $\mathcal{I}(l) = \mathcal{I}(\mathsf{z3.Length}(e'_1)) = \mathsf{len}(e'_4) = l'$. Therefore,

$$
\begin{aligned}
\mathcal{I}([\![\mathsf{SubStr}(E_1, E_2, E_3)]\!]_{\mathcal{S},\Gamma,\mathcal{T}}) &= \mathcal{I}(\mathsf{ite}(e_1 = \mathrm{Null} \vee \mathsf{IsStr}(e_2) \vee \mathsf{IsStr}(e_3), \mathrm{Null}, s)) \\
&= \mathsf{ite}(\mathcal{I}(e_1) = \mathrm{Null} \vee \mathcal{I}(\mathsf{IsStr}(e_2)) \vee \mathcal{I}(\mathsf{IsStr}(e_3)), \\
&\qquad \mathrm{Null}, \mathcal{I}(s)) \\
&= \mathsf{ite}(e_4 = \mathrm{Null} \vee \mathsf{IsStr}(e_5) \vee \mathsf{IsStr}(e_6)), s') \\
&= [\![\mathsf{SubStr}(E_1, E_2, E_3)]\!]_{\mathcal{I}(\Gamma),\mathcal{I}(\mathcal{T})}
\end{aligned}
$$

13. Inductive case: $\phi = \mathsf{Concate}(E_1, E_2)$.

$[\![\mathsf{Concate}(E_1, E_2)]\!]_{\mathcal{S},\Gamma,\mathcal{T}} = \mathsf{ite}(v_1 = \mathrm{Null} \vee v_2 = \mathrm{Null}, \bot, \mathsf{z3.Concat}(v_1, v_2))$ where $v_1 = [\![\mathsf{ToStr}(E_1)]\!]_{\mathcal{S},\Gamma,\mathcal{T}}$ and $v_2 = [\![\mathsf{ToStr}(E_2)]\!]_{\mathcal{S},\Gamma,\mathcal{T}}$ by Figure 18. $[\![\mathsf{Concate}(E_1, E_2)]\!]_{\mathcal{I}(\Gamma),\mathcal{I}(\mathcal{T})} = \mathsf{ite}(v'_1 = \mathrm{Null} \vee v'_2 = \mathrm{Null}, \bot, \mathsf{z3.Concat}(v'_1, v'_2))$ where

$v_1' = [\![\mathsf{ToStr}(E_1)]\!]_{\mathcal{I}(\Gamma),\mathcal{I}(\mathcal{T})}$ and $v_2' = [\![\mathsf{ToStr}(E_2)]\!]_{\mathcal{I}(\Gamma),\mathcal{I}(\mathcal{T})}$ by Figure 17. By inductive hypothesis, we have $\mathcal{I}(v_1) = \mathcal{I}([\![E_1]\!]_{\mathcal{S},\Gamma,\mathcal{T}}) = [\![E_1]\!]_{\mathcal{I}(\Gamma),\mathcal{I}(\mathcal{T})} = v_1'$ and $\mathcal{I}(v_2) = \mathcal{I}([\![E_2]\!]_{\mathcal{S},\Gamma,\mathcal{T}}) = [\![E_2]\!]_{\mathcal{I}(\Gamma),\mathcal{I}(\mathcal{T})} = v_2'$. Furthermore, by the semantics of z3.Concat, $\mathcal{I}(\mathsf{z3.Concat}) = \mathsf{Concate}$. Therefore,

$$
\begin{aligned}
\mathcal{I}([\![\mathsf{Concate}(E_1, E_2)]\!]_{\mathcal{S},\Gamma,\mathcal{T}}) &= \mathcal{I}(\mathsf{ite}(v_1 = \mathsf{Null} \vee v_2 = \mathsf{Null}, \bot, \mathsf{z3.Concat}(v_1, v_2))) \\
&= \mathsf{ite}(\mathcal{I}(v_1) = \mathsf{Null} \vee \mathcal{I}(v_2) = \mathsf{Null}, \bot, \\
&\qquad \mathcal{I}(\mathsf{z3.Concat})(\mathcal{I}(v_1), \mathcal{I}(v_2))) \\
&= \mathsf{ite}(v_1' = \mathsf{Null} \vee v_2' = \mathsf{Null}, \bot, \mathsf{Concate}(v_1', v_2')) \\
&= [\![\mathsf{Concate}(E_1, E_2)]\!]_{\mathcal{I}(\Gamma),\mathcal{I}(\mathcal{T})}
\end{aligned}
$$

14. Inductive case: $E = \mathsf{Strftime}(\kappa, E)$.

$[\![\mathsf{Strftime}(\kappa, E)]\!]_{\mathcal{S},\Gamma,\mathcal{T}} = \mathsf{ite}(v = \mathsf{Null}, \mathsf{Null}, \mathsf{ite}(\kappa = \text{``\%Y''}, v[0], \mathsf{ite}(\kappa = \text{``\%M''}, v[1], v[2])))$ where $v = [\![E]\!]_{\mathcal{S},\Gamma,\mathcal{T}}$ by Figure 18. $[\![\mathsf{Strftime}(\kappa, E)]\!]_{\mathcal{I}(\Gamma),\mathcal{I}(\mathcal{T})} = \mathsf{ite}(v = \mathsf{Null}, \mathsf{Null}, \mathsf{ite}(\kappa = \text{``\%Y''}, v[0], \mathsf{ite}(\kappa = \text{``\%M''}, v[1], v[2])))$ where $v = [\![E]\!]_{\mathcal{I}(\Gamma),\mathcal{I}(\mathcal{T})}$ by Figure 17. By inductive hypothesis, we have $\mathcal{I}(v) = \mathcal{I}([\![E]\!]_{\mathcal{S},\Gamma,\mathcal{T}}) = [\![E]\!]_{\mathcal{I}(\Gamma),\mathcal{I}(\mathcal{T})} = v'$. Therefore,

$$
\begin{aligned}
\mathcal{I}([\![\mathsf{Strftime}(\kappa, E)]\!]_{\mathcal{S},\Gamma,\mathcal{T}}) &= \mathcal{I}(\mathsf{ite}(v = \mathsf{Null}, \mathsf{Null}, \mathsf{ite}(\kappa = \text{``\%Y''}, v[0], \\
&\qquad \mathsf{ite}(\kappa = \text{``\%M''}, v[1], v[2])))) \\
&= \mathsf{ite}(\mathcal{I}(v) = \mathsf{Null}, \mathsf{Null}, \mathsf{ite}(\kappa = \text{``\%Y''}, \mathcal{I}(v)[0], \\
&\qquad \mathsf{ite}(\kappa = \text{``\%M''}, \mathcal{I}(v)[1], \mathcal{I}(v)[2]))) \\
&= \mathsf{ite}(v' = \mathsf{Null}, \mathsf{Null}, \mathsf{ite}(\kappa = \text{``\%Y''}, v'[0], \\
&\qquad \mathsf{ite}(\kappa = \text{``\%M''}, v'[1], v'[2]))) \\
&= [\![\mathsf{Strftime}(\kappa, E)]\!]_{\mathcal{I}(\Gamma),\mathcal{I}(\mathcal{T})}
\end{aligned}
$$

15. Inductive case: $E = \mathsf{JulianDay}(E)$.

$[\![\mathsf{JulianDay}(E)]\!]_{\mathcal{S},\Gamma,\mathcal{T}} = \mathsf{ToJulianDay}(v)$ where $v = [\![E]\!]_{\mathcal{S},\Gamma,\mathcal{T}}$ if $v$ is evaluated to be a date by Figure 18. Also, $\mathsf{ToJulianDay}(v) = \lfloor 365.25 * (y + 4716) \rfloor + \lfloor 30.6001 * (m + 4716) \rfloor + d + c - 1524.5$ where $y = v[1] \leq 2? v[0] - 1 : v[0]$, $m = v[1] \leq 2? v[1] + 12 : v[1]$, $d = v[2]$, and $c = 2 - \lfloor y/100 \rfloor + \lfloor y/400 \rfloor$. $[\![\mathsf{JulianDay}(E)]\!]_{\mathcal{I}(\Gamma),\mathcal{I}(\mathcal{T})} = a_1 + a_2 + d' + c' - 1524.5$ where $v' = [\![E]\!]_{\mathcal{I}(\Gamma),\mathcal{I}(\mathcal{T})}$, $y' = \mathsf{ite}(v'[1] \leq 2, v'[0] - 1, v'[0])$, $m' = \mathsf{ite}(v'[1] \leq 2, v'[1] + 12, v'[1])$, $d' = v'[2]$, $c' = 2 - \mathsf{fdiv}(y', 100) + \mathsf{fdiv}(y', 400)$, $a_1 = \mathsf{fdiv}(36525 * (y' + 4716), 10^2)$, and $a_2 = \mathsf{fdiv}(306001 * (m' + 1), 10^4)$ if $v_1$ is evaluated to be a date by Figure 17. By inductive hypothesis, we have $\mathcal{I}(v) = \mathcal{I}([\![E]\!]_{\mathcal{S},\Gamma,\mathcal{T}}) = [\![E]\!]_{\mathcal{I}(\Gamma),\mathcal{I}(\mathcal{T})} = v'$. Furthermore, by the semantics of fdiv, $\mathcal{I}(\lfloor 365.25 * (y + 4716) \rfloor) = a_1$ and $\mathcal{I}(\lfloor 30.6001 * (m + 4716) \rfloor) = a_2$. Therefore,

$$
\begin{aligned}
\mathcal{I}([\![\mathsf{JulianDay}(E)]\!]_{\mathcal{S},\Gamma,\mathcal{T}}) &= \mathcal{I}(\mathsf{ToJulianDay}(v)) \\
&= \mathcal{I}(\lfloor 365.25 * (y + 4716) \rfloor + \lfloor 30.6001 * (m + 4716) \rfloor \\
&\qquad + d + c - 1524.5) \\
&= a_1 + a_2 + d' + c' - 1524.5 \\
&= [\![\mathsf{JulianDay}(E)]\!]_{\mathcal{I}(\Gamma),\mathcal{I}(\mathcal{T})}
\end{aligned}
$$

16. Inductive case: $E = \mathsf{DateShift}(E, i, \delta)$.

$[\![\mathsf{DateShift}(E, i, \delta)]\!]_{\mathcal{S},\Gamma,\mathcal{T}} = \mathsf{DateAdd}(v, i, \delta)$ where $v = [\![\mathsf{ToDate}(E)]\!]_{\mathcal{S},\Gamma,\mathcal{T}}$ if $v$ is evaluated to be a date by Figure 18. Also, $\mathsf{DateAdd}(v, i, \delta)$ is defined as follows:

(a) If $\delta = \text{``Year''}$, then $\mathsf{DateAdd}(v, i, \delta) = \mathsf{ite}(v'[0] < \mathsf{MIN\_YEAR} \vee \mathsf{MAX\_YEAR} < v'[0], \mathsf{Null}, \mathsf{Null}, v')$ where $v' = [v[0] + i, v[1], v[2]]$ as $i$ can be negative and dates falling outside the valid date range are regarded as Null.

(b) If $\delta = \text{``Month''}$, then $\mathsf{DateAdd}(v, i, \delta) = \mathsf{ite}(v'[0] < \mathsf{MIN\_YEAR} \vee \mathsf{MAX\_YEAR} < v'[0], \mathsf{Null}, \mathsf{Null}, v')$ where $v' = [v[0] + \mathsf{fdiv}(v[1] + i, 12), (v[1] + i)\%12, v[2]]$.

(c) If $\delta = \text{``Day''}$, then $\mathsf{DateAdd}(v, i, \delta) = \mathsf{ite}(v' < \mathsf{MIN\_DATE} \wedge v' > \mathsf{MAX\_DATE}, \mathsf{Null}, v')$ where $v'$ is a new data variable s.t. $\mathsf{SinceBegin}(v') - \mathsf{SinceBegin}(v) = i$. In addition, the SinceBegin function counts the ordinal number of a date from a certain day, e.g., "0000-01-01", which can be defined as

$$\text{SinceBegin}(y, m, d) = \text{year2day}(y) + \text{month2day}(m) + d \text{ where}$$

$$
\begin{aligned}
\text{year2day}(y) &= 365 \times y - 1 + \text{fdiv}(y - 1, 4) - \text{fdiv}(y - 1, 100) + \text{fdiv}(y - 1, 400) \\
\text{month2day}(m) &= \textstyle\sum_{i=1}^{m-1} \text{ite}(i \in \{1, 3, 5, 7, 8, 10, 12\}, 31, \\
&\qquad \text{ite}(i = 2, 28 + \text{ite}(\text{leap}(y), 1, 0), 30))
\end{aligned}
$$

. Thus, $\text{SinceBegin}(v') - \text{SinceBegin}(v) = i$ ensure the date $v'$ is $i$ days away from the date $v$.

$\llbracket \text{DateShift}(E) \rrbracket_{\mathcal{I}(\Gamma), \mathcal{I}(\mathcal{T})} = \text{ite}(\delta = \text{"Year"}, \text{DateShiftByYears}(v, i), \text{ite}(\delta = \text{"Month"}, \text{DateShiftByMonths}(v, i), \text{DateShiftByDays}(v, i)))$ where $v_1 = \llbracket E \rrbracket_{\mathcal{I}(\Gamma), \mathcal{I}(\mathcal{T})}$ if $v_1$ is evaluated to be a date by Figure 17. By the semantics of the DateShiftByYears, DateShiftByMonths and DateShiftByDays functions, they corresponding to the case 16a, 16b and 16c. Therefore,

$$
\begin{aligned}
\mathcal{I}(\llbracket \text{DateShift}(E, i, \delta) \rrbracket_{\mathcal{S}, \Gamma, \mathcal{T}}) &= \mathcal{I}(\text{ite}(\delta = \text{"Year"}, \text{DateShiftByYears}(v, i), \\
&\qquad \text{ite}(\delta = \text{"Month"}, \text{DateShiftByMonths}(v, i), \\
&\qquad \text{DateShiftByDays}(v, i)))) \\
&= \llbracket \text{DateShift}(E, i, \delta) \rrbracket_{\mathcal{I}(\Gamma), \mathcal{I}(\mathcal{T})}
\end{aligned}
$$

$\square$

**Theorem 3** (Correctness of predicate encoding). *Let $D$ be a database over schema $\mathcal{S}$, $xs$ be a tuple list, and $\phi$ be a predicate. Consider a symbolic database $\Gamma$ over $\mathcal{S}$, a list of symbolic tuples $\mathcal{T}$, and $\phi$'s symbolic encoding $\llbracket \phi \rrbracket_{\mathcal{S}, \Gamma, \mathcal{T}}$. For any satisfying interpretation $\mathcal{I}$ with $\mathcal{I}(\Gamma) = D \wedge \mathcal{I}(\mathcal{T}) = xs$, evaluating $\phi$ over the database $D$ and the tuple list $xs$ yields the interpretation of $\phi$'s symbolic encoding $\mathcal{I}(\llbracket \phi \rrbracket_{\mathcal{S}, \Gamma, \mathcal{T}})$, i.e.,*

$$\mathcal{I}(\Gamma) = D \wedge \mathcal{I}(\mathcal{T}) = xs \Rightarrow \llbracket \phi \rrbracket_{D, xs} = \mathcal{I}(\llbracket \phi \rrbracket_{\mathcal{S}, \Gamma, \mathcal{T}})$$

**Lemma 2.** *Suppose $\llbracket \phi \rrbracket_{D, xs}$ is valid, then $\mathcal{I}(\Gamma) = D \wedge \mathcal{I}(\mathcal{T}) = xs \Rightarrow \llbracket \phi \rrbracket_{\mathcal{I}(\Gamma), \mathcal{I}(\mathcal{T})} = \mathcal{I}(\llbracket \phi \rrbracket_{\mathcal{S}, \Gamma, \mathcal{T}})$ holds.*

*Proof.* Theorem 3 is proved by proving Lemma 2. By structural induction on $\phi$.

1. Base cases and some inductive cases are proved in He et al. (2024).

2. Inductive case: $\phi = \text{PrefixOf}(s, E)$.

   $\llbracket \text{PrefixOf}(s, E) \rrbracket_{\mathcal{S}, \Gamma, \mathcal{T}} = \text{ite}(v = \text{Null}, \text{Null}, \text{z3.PrefixOf}(s, v))$ where $v = \llbracket \text{ToStr}(E) \rrbracket_{\mathcal{S}, \Gamma, \mathcal{T}}$ by Figure 19. $\llbracket \text{PrefixOf}(s, E) \rrbracket_{\mathcal{I}(\Gamma), \mathcal{I}(\mathcal{T})} = \text{ite}(v' = \text{Null}, \text{Null}, \text{PrefixOf}(s, v'))$ where $v' = \llbracket \text{ToStr}(E) \rrbracket_{\mathcal{I}(\Gamma), \mathcal{I}(\mathcal{T})}$ by Figure 17. By inductive hypothesis, we have $\mathcal{I}(v) = \mathcal{I}(\llbracket \text{ToStr}(E) \rrbracket_{\mathcal{I}(\Gamma), \mathcal{I}(\mathcal{T})}) = \llbracket \text{ToStr}(E) \rrbracket_{\mathcal{S}, \Gamma, \mathcal{T}} = v'$. Furthermore, since the Z3 builtin function z3.PrefixOf precisely captures the semantics of PrefixOf, we have $\mathcal{I}(\text{z3.PrefixOf}(s, v)) = \mathcal{I}(\text{z3.PrefixOf}(s, \llbracket \text{ToStr}(E) \rrbracket_{\mathcal{S}, \Gamma, \mathcal{T}})) = \text{PrefixOf}(s, \mathcal{I}(\llbracket \text{ToStr}(E) \rrbracket_{\mathcal{S}, \Gamma, \mathcal{T}})) = \text{PrefixOf}(s, \llbracket \text{ToStr}(E) \rrbracket_{\mathcal{I}(\Gamma), \mathcal{I}(\mathcal{T})}) = \text{PrefixOf}(s, v')$. Therefore,

   $$
   \begin{aligned}
   \mathcal{I}(\llbracket \text{PrefixOf}(s, E) \rrbracket_{\mathcal{S}, \Gamma, \mathcal{T}}) &= \mathcal{I}(\text{ite}(v = \text{Null}, \text{Null}, \text{z3.PrefixOf}(s, v))) \\
   &= \text{ite}(\mathcal{I}(v) = \text{Null}, \text{Null}, \mathcal{I}(\text{z3.PrefixOf}(s, v))) \\
   &= \text{ite}(v' = \text{Null}, \text{Null}, \text{PrefixOf}(s, v')) \\
   &= \llbracket \text{PrefixOf}(s, E) \rrbracket_{\mathcal{I}(\Gamma), \mathcal{I}(\mathcal{T})}
   \end{aligned}
   $$

3. Inductive case: $\phi = \text{SuffixOf}(s, E)$.

   $\llbracket \text{SuffixOf}(s, E) \rrbracket_{\mathcal{S}, \Gamma, \mathcal{T}} = \text{ite}(v = \text{Null}, \text{Null}, \text{z3.SuffixOf}(s, v))$ where $v = \llbracket \text{ToStr}(E) \rrbracket_{\mathcal{S}, \Gamma, \mathcal{T}}$ by Figure 19. $\llbracket \text{SuffixOf}(s, E) \rrbracket_{\mathcal{I}(\Gamma), \mathcal{I}(\mathcal{T})} = \text{ite}(v' = \text{Null}, \text{Null}, \text{SuffixOf}(s, v'))$ where $v' = \llbracket \text{ToStr}(E) \rrbracket_{\mathcal{I}(\Gamma), \mathcal{I}(\mathcal{T})}$ by Figure 17. By inductive hypothesis, we have $\mathcal{I}(v) = \mathcal{I}(\llbracket \text{ToStr}(E) \rrbracket_{\mathcal{I}(\Gamma), \mathcal{I}(\mathcal{T})}) = \llbracket \text{ToStr}(E) \rrbracket_{\mathcal{S}, \Gamma, \mathcal{T}} = v'$. Furthermore, since the Z3 builtin function z3.SuffixOf precisely captures the semantics of SuffixOf, we have $\mathcal{I}(\text{z3.SuffixOf}(s, v)) = \mathcal{I}(\text{z3.SuffixOf}(s, \llbracket \text{ToStr}(E) \rrbracket_{\mathcal{S}, \Gamma, \mathcal{T}})) =$

$\text{SuffixOf}(s, \mathcal{I}(\llbracket\text{ToStr}(E)\rrbracket_{\mathcal{S},\Gamma,\mathcal{T}})) = \text{SuffixOf}(s, \llbracket\text{ToStr}(E)\rrbracket_{\mathcal{I}(\Gamma),\mathcal{I}(\mathcal{T})}) = \text{SuffixOf}(s, v')$ . Therefore,

$$
\begin{aligned}
\mathcal{I}(\llbracket\text{SuffixOf}(s, E)\rrbracket_{\mathcal{S},\Gamma,\mathcal{T}}) &= \mathcal{I}(\text{ite}(v = \text{Null}, \text{Null}, \text{z3.SuffixOf}(s, v))) \\
&= \text{ite}(\mathcal{I}(v) = \text{Null}, \text{Null}, \mathcal{I}(\text{z3.SuffixOf}(s, v))) \\
&= \text{ite}(v' = \text{Null}, \text{Null}, \text{SuffixOf}(s, v')) \\
&= \llbracket\text{SuffixOf}(s, E)\rrbracket_{\mathcal{I}(\Gamma),\mathcal{I}(\mathcal{T})}
\end{aligned}
$$

4. Inductive case: $\phi = \text{Like}(s, E)$.

$\llbracket\text{Like}(s, E)\rrbracket_{\mathcal{S},\Gamma,\mathcal{T}} = \text{ite}(v = \text{Null}, \bot, \text{z3.RegexMatch}(s))$ where $v = \llbracket\text{ToStr}(E)\rrbracket_{\mathcal{S},\Gamma,\mathcal{T}}$ by Figure 19. $\llbracket\text{Like}(s, E)\rrbracket_{\mathcal{I}(\Gamma),\mathcal{I}(\mathcal{T})} = \text{ite}(v' = \text{Null}, \text{Null}, \text{RegexMatch}(s, v'))$ where $v' = \llbracket\text{ToStr}(E)\rrbracket_{\mathcal{I}(\Gamma),\mathcal{I}(\mathcal{T})}$ by Figure 17. By inductive hypothesis, we have $\mathcal{I}(v) = \mathcal{I}(\llbracket\text{ToStr}(E)\rrbracket_{\mathcal{S},\Gamma,\mathcal{T}}) = \llbracket\text{ToStr}(E)\rrbracket_{\mathcal{I}(\Gamma),\mathcal{I}(\mathcal{T})} = v'$. Furthermore, since Z3 precisely support regular expressions, we have $\mathcal{I}(\text{z3.RegexMatch}(s, v)) = \mathcal{I}(\text{z3.RegexMatch}(s, \llbracket\text{ToStr}(E)\rrbracket_{\mathcal{S},\Gamma,\mathcal{T}})) = \text{RegexMatch}(s, \mathcal{I}(\llbracket\text{ToStr}(E)\rrbracket_{\mathcal{S},\Gamma,\mathcal{T}})) = \text{RegexMatch}(s, \llbracket\text{ToStr}(E)\rrbracket_{\mathcal{I}(\Gamma),\mathcal{I}(\mathcal{T})}) = \text{RegexMatch}(s, v')$. Therefore,

$$
\begin{aligned}
\mathcal{I}(\llbracket\text{Like}(s, E)\rrbracket_{\mathcal{S},\Gamma,\mathcal{T}}) &= \mathcal{I}(\text{ite}(v = \text{Null}, \bot, \text{z3.RegexMatch}(s, v))) \\
&= \text{ite}(\mathcal{I}(v) = \text{Null}, \bot, \mathcal{I}(\text{z3.RegexMatch}(s, v))) \\
&= \text{ite}(v' = \text{Null}, \bot, \text{RegexMatch}(s, v')) \\
&= \llbracket\text{Like}(s, E)\rrbracket_{\mathcal{I}(\Gamma),\mathcal{I}(\mathcal{T})}
\end{aligned}
$$

5. Inductive case: $\phi = \text{Contain}(s, E)$.

$\llbracket\text{Contain}(s, E)\rrbracket_{\mathcal{S},\Gamma,\mathcal{T}} = \text{ite}(v = \text{Null}, \bot, \text{z3.RegexMatch}(s', v))$ where $s' = \text{Concate}(\text{".*"}, s, \text{".*"})$ and $v = \llbracket\text{ToStr}(E)\rrbracket_{\mathcal{S},\Gamma,\mathcal{T}}$ by Figure 19. $\llbracket\text{Contain}(s, E)\rrbracket_{\mathcal{I}(\Gamma),\mathcal{I}(\mathcal{T})} = \llbracket\text{Like}(s'', E)\rrbracket_{\mathcal{I}(\Gamma),\mathcal{I}(\mathcal{T})} = \text{ite}(v' = \text{Null}, \text{Null}, \text{RegexMatch}(s'', v'))$ where $s'' = \text{Concate}(\text{"%"}, s, \text{"%"})$ and $v' = \llbracket\text{ToStr}(E)\rrbracket_{\mathcal{I}(\Gamma),\mathcal{I}(\mathcal{T})}$ by Figure 17. Futhermore, by the semantics of z3.RegexMatch and RegexMatch, we know $s'$ and $s''$ represent the same regular expression, and $\mathcal{I}(\text{z3.RegexMatch}(s', x')) = \text{RegexMatch}(s'', x'')$ iff $\mathcal{I}(x') = x''$. Therefore,

$$
\begin{aligned}
\mathcal{I}(\llbracket\text{Contain}(s, E)\rrbracket_{\mathcal{S},\Gamma,\mathcal{T}}) &= \mathcal{I}(\text{ite}(v = \text{Null}, \bot, \text{z3.RegexMatch}(s', v))) \\
&= \text{ite}(\mathcal{I}(v) = \text{Null}, \bot, \mathcal{I}(\text{z3.RegexMatch}(s', v))) \\
&= \text{ite}(v' = \text{Null}, \bot, \text{RegexMatch}(s'', v')) \\
&= \llbracket\text{Like}(s'', E)\rrbracket_{\mathcal{I}(\Gamma),\mathcal{I}(\mathcal{T})} \\
&= \llbracket\text{Contain}(s, E)\rrbracket_{\mathcal{I}(\Gamma),\mathcal{I}(\mathcal{T})}
\end{aligned}
$$

6. Inductive case: $\phi = E_1 \odot E_2$.

$\llbracket E_1 \odot E_2\rrbracket_{\mathcal{S},\Gamma,\mathcal{T}} = \text{ite}(v_1 = \text{Null} \vee v_2 = \text{Null}, \bot, v_1 \odot v_2)$ where $v_1 = \llbracket E_1\rrbracket_{\mathcal{S},\Gamma,\mathcal{T}}$ and $v_2 = \llbracket E_2\rrbracket_{\mathcal{S},\Gamma,\mathcal{T}}$ if $v_1$ and $v_2$ share the same type, i.e., $\text{Type}(v_1) = \text{Type}(v_2)$ by Figure 19. $\llbracket E_1 \odot E_2\rrbracket_{\mathcal{I}(\Gamma),\mathcal{I}(\mathcal{T})} = \text{ite}(v_1' = \text{Null} \vee v_2' = \text{Null}, \bot, v_1' \odot v_2')$ where $v_1 = \llbracket E_1\rrbracket_{\mathcal{I}(\Gamma),\mathcal{I}(\mathcal{T})}$ and $v_2' = \llbracket E_2\rrbracket_{\mathcal{I}(\Gamma),\mathcal{I}(\mathcal{T})}$ if $v_1'$ and $v_2'$ share the same type, i.e., $\text{Type}(v_1') = \text{Type}(v_2')$ by Figure 17. Note that this operation only works for $E_1$ and $E_2$ sharing the same type which is consistent with MYSQL. By inductive hypothesis, we have $\mathcal{I}(v_1) = \mathcal{I}(\llbracket E_1\rrbracket_{\mathcal{I}(\Gamma),\mathcal{I}(\mathcal{T})}) = \llbracket E_1\rrbracket_{\mathcal{S},\Gamma,\mathcal{T}} = v_1'$ and $\mathcal{I}(v_2) = \mathcal{I}(\llbracket E_2\rrbracket_{\mathcal{I}(\Gamma),\mathcal{I}(\mathcal{T})}) = \llbracket E_2\rrbracket_{\mathcal{S},\Gamma,\mathcal{T}} = v_2'$. Therefore, when $E_1$ and $E_2$ have the same type, we have

$$
\begin{aligned}
\mathcal{I}(\llbracket E_1 \odot E_2\rrbracket_{\mathcal{S},\Gamma,\mathcal{T}}) &= \mathcal{I}(\text{ite}(v_1 = \text{Null} \vee v_2 = \text{Null}, \bot, v_1 \odot v_2)) \\
&= \text{ite}(\mathcal{I}(v_1) = \text{Null} \vee \mathcal{I}(v_2) = \text{Null}, \bot, \mathcal{I}(v_1) \odot \mathcal{I}(v_2)) \\
&= \text{ite}(v_1' = \text{Null} \vee v_2' = \text{Null}, \bot, v_1' \odot v_2') \\
&= \llbracket E_1 \odot E_2\rrbracket_{\mathcal{I}(\Gamma),\mathcal{I}(\mathcal{T})}
\end{aligned}
$$

$\square$

**Theorem 2** (Equivalence under set semantics). *Given two relations $R_1 = [t_1, \ldots, t_n]$ and $R_2 = [r_1, \ldots, r_m]$, if formula (2) is valid, then $R_1$ and $R_2$ are equivalent under set semantics.*

*Proof.* Let $F_1$ be the first conjunct of formula (2), i.e., $\bigwedge_{i=1}^{n}(\neg\text{Del}(t_i) \rightarrow \vee_{j=1}^{m}(\neg\text{Del}(r_j) \wedge t_i = r_j)$, and let $F_2$ be the second conjunct of formula (2), i.e., $\bigwedge_{j=1}^{m}(\neg\text{Del}(r_j) \rightarrow \vee_{i=1}^{n}(\neg\text{Del}(t_i) \wedge r_j = t_i)$. Since formula (2) is valid, both $F_1$ and $F_2$ are valid. Now consider $F_1$. It specifies for any tuple $t_i \in R_1$, if $t_i$ is not deleted, then there exists a tuple $r_j$ that is not deleted and $t_i = r_j$. By the definition of $\subseteq$, $R_1 \subseteq R_2$. Similarly, $F_2$ specifies $R_2 \subseteq R_1$. Therefore, $R_1 = R_2$.

$\square$

