# OpenReview forum: "SpotIt: Evaluating Text-to-SQL Evaluation with Formal Verification"
_ICLR.cc/2026/Conference — ICLR 2026 Poster_

### Official Review · Reviewer_p5hF · 2025-10-26

**Soundness:** 3
**Presentation:** 3
**Contribution:** 3
**Rating:** 4
**Confidence:** 5

**Summary:**

This paper introduces SPOTIT, a framework for improving the evaluation of Text-to-SQL systems via formal verification. Instead of relying solely on benchmark-provided test cases, SPOTIT aims to automatically generate counterexample databases that can distinguish between a generated SQL query and the corresponding gold (ground-truth) query.

SPOTIT extends VeriEQL (He et al., 2024) to handle a broader range of SQL constructs frequently used in real-world datasets such as BIRD, including string and date operations (strftime, JulianDay, ToStr, substr, LIKE, etc.). The system is applied to outputs from ten Text-to-SQL models, revealing that many incorrect or semantically mismatched queries pass traditional dataset-based evaluations but are correctly identified as incorrect by SPOTIT.

**Strengths:**

* **S1- Clear motivation and formulation**: The paper identifies a well-motivated gap in evaluating Text-to-SQL systems due to incomplete or non-discriminative test databases and formalizes the problem elegantly.

* **S2- Extension of formal verification**: Extending VeriEQL to cover a richer subset of SQL operators is a nontrivial and practically useful engineering contribution.

* **S3- Comprehensive evaluation**: Applying the verifier to ten different Text-to-SQL models on BIRD demonstrates empirical value.

* **S4- Readable and well-structured**: The paper includes illustrative examples that make the approach and findings easy to follow.

**Weaknesses:**

* **W1- Limited related work coverage**:
  The paper omits prior research on test data generation and SQL verification, including approaches based on constraint solving and fuzzing [a,b,c].

  * [a] Chandra et al. Data generation for testing and grading SQL queries. VLDB Journal, 2015.

  * [b] Somwase et al. Data Generation for Testing Complex Queries. arXiv, 2024.

  * [c] Zhong et al. Semantic Evaluation for Text-to-SQL with Distilled Test Suites. EMNLP, 2020.

* **W2- Context and positioning**:
  Related work is moved to the appendix, which weakens the contextual framing. A short related work section in the main text would help readers understand how SPOTIT builds upon and differs from existing formal or data-driven SQL evaluation approaches.

* **W3- Limited scope of expressivity**:
While the supported SQL subset has been expanded, the framework still cannot handle aggregation subtleties (e.g., nested subqueries, HAVING clauses, or complex joins) as far as can be inferred from Fig 2. Clarifying this limitation would improve transparency.

* **W4- Scalability concerns**:
The reliance on a SAT solver introduces potential scalability issues as query and schema complexity increase. The paper would benefit from runtime statistics or analysis of computational overhead.

**Questions:**

1. How is the value of K (the search bound) determined? Larger K improves completeness but increases runtime—what trade-offs were observed, and what values were used in experiments?

2. Which query classes cannot be tested using bounded equivalence? Are such queries present in benchmarks such as BIRD?

3. How does runtime scale with the number of tables, query complexity, and schema size?

---

> ### Author Response · Authors · 2025-11-22
> **Thank you for your feedback!**
>
> Please kindly refer to our response to all for the effect of K, an evaluation on Spider 2.0 benchmarks, and clarification on the SQL features we currently can and cannot support.
>
> > Prior research on test data generation and SQL verification.
>
> Thank you for pointing out the related work. Test data generation can be very useful for detecting surface-level query non-equivalence. Indeed, BIRD and Spider 2.0 already use complex database instances to test query equivalence and can filter out many incorrect answers. However, when the counterexamples require very specific structures (which is the case for many query pairs that passed EX-test but failed SpotIt as seen in App. D), random fuzzing/testing becomes less reliable for refuting equivalence. In contrast, SpotIt *systematically* searches over the space of possible differentiating databases, finds minimal counter-examples, and provides a *formal guarantee*: if SpotIt deems two SQL queries equivalent, then no counterexample of size less than a fixed number exists.
>
> Computational resources permitted, one could in principle run a portfolio of test-based and verification-based equivalence-checking methods in parallel to more quickly detect non-equivalence. This is an orthogonal but interesting future direction.
>
> We have incorporated this discussion into the paper.
>
>
> > A short related work section in the main text.
>
> Thank you for this suggestion. We have added a short related work section in the main text, as the revision and the final version allow one additional page.
>
>
> >  Reliance on a SAT solver. The paper would benefit from runtime statistics or analysis of computational overhead.
>
> We would like to point out that the reliance on an SMT solver does not necessarily imply a lack of practicality. SAT/SMT solvers are being routinely used in industrial applications (e.g., [1]).  We believe the fact that SMT solvers already scale to existing high-profile benchmarks is itself a meaningful finding for the Text-to-SQL community.
>
> We reported the runtime of SpotIt in Table 2, which shows that the average runtime is under 4 seconds for all Text-to-SQL methods we evaluated.
>
> [1] “A billion SMT queries a day.” Amazon. 2022. https://www.amazon.science/blog/a-billion-smt-queries-a-day
>
> > Which query classes cannot be tested using bounded equivalence? Are such queries present in benchmarks such as BIRD?
>
> Thank you for the feedback. Please refer to our answers in the response to all regarding the features we currently can and cannot support. Table 2 of the paper reports the percentage of supported SQL query pairs by SpotIt, which ranges from 93.89% to 97.17%.
>
>
> > How does runtime scale with the number of tables, query complexity, and schema size?
>
> We perform such an analysis and report how the runtime changes with respect to the number of columns, integrity constraints, tables, sub-queries in the gold SQL, and nodes in the abstract syntax tree in the gold SQL. We found that all parameters except for the number of tables are positively correlated with median runtime.
>
> | # columns | Median runtime (s) |
> |-------------|---------------------|
> | 11          | 0.4310              |
> | 21          | 0.1701              |
> | 31          | 0.1963              |
> | 48          | 0.3887              |
> | 55          | 0.5576              |
> | 64          | 0.3007              |
> | 71          | 0.5365              |
> | 89          | 0.2672              |
> | 94          | 0.3154              |
> | 115         | 0.6537              |
> | 199         | 0.8006              |
>
> | # constraints | Median runtime (s) |
> |------------------|---------------------|
> | 5                | 0.2842              |
> | 7                | 0.1701              |
> | 10               | 0.4324              |
> | 16               | 0.5576              |
> | 17               | 0.3887              |
> | 19               | 0.1963              |
> | 21               | 0.5365              |
> | 36               | 0.6877              |
>
> | # tables | Median runtime (s) |
> |-------------|---------------------|
> | 3           | 0.2842              |
> | 4           | 0.4310              |
> | 5           | 0.1701              |
> | 6           | 0.6537              |
> | 7           | 0.8006              |
> | 8           | 0.4249              |
> | 10          | 0.1963              |
> | 13          | 0.3154              |
>
> | # subqueries | Median runtime (s) |
> |----------------|---------------------|
> | 0              | 0.3676              |
> | 1              | 0.3972              |
> | 2              | 1.9128              |
>
>
> | # AST nodes | Median runtime (s) |
> |------------|---------------------|
> | 0–19       | 0.1812              |
> | 20–39      | 0.3408              |
> | 40–59      | 0.3426              |
> | 60–79      | 0.4216              |
> | 80–99      | 0.9683              |
> | 100–119    | 1.1865              |
> | 120–139    | 0.2053              |
> | 140–159    | 1.2395              |

---

> > ### Author Response · Authors · 2025-11-26
> >
> > Dear reviewer, we hope our responses address your questions and concerns. Please let us know if there is anything else you would like us to clarify. Thank you!
> >
> > Sincerely,
> >
> > Authors

---

> > ### Comment · Reviewer_p5hF · 2025-11-26
> > **Follow-up on rebuttal**
> >
> > I thank the authors for their response. However, several concerns remain insufficiently addressed:
> >
> > **W2 – Context and Positioning of Contributions**
> >
> > Prior work on synthetic database generation follows a workflow highly similar to SpotIt: (1) generate mutated queries or logical neighbors P of a gold query Q, and (2) construct databases or counterexamples that distinguish Q from P. Several of these works also incorporate SMT solvers to address essentially the same problem. While the revised version adds those missing references, it is still unclear how SpotIt advances the state of the art.
> > Please clarify:
> > - In what ways does your search for  P and for counterexample databases differ from prior approaches, and why are these differences significant?
> > - Is the main contribution primarily an extension of VeriEQL to support strings and date operations, or is there a deeper conceptual novelty?
> >
> > **Query Classes Not Captured by Bounded Equivalence**
> >
> > The limitations of bounded equivalence remain unclear. Why exactly is the bound k a limiting factor for recursive queries or window functions? It also seems that output size influences equivalence: for example, you noted that the following two queries are equivalent under k≤5 but not under k=6.
> >
> > - Gold: SELECT T1.full_name FROM superhero AS T1 INNER JOIN race AS T2 ON T1.race_id = T2.id WHERE T2.race = 'Demi-God'
> >
> > - Prediction: SELECT s.full_name FROM superhero s JOIN race r ON s.race_id = r.id WHERE r.race = 'Demi-God' LIMIT 5

---

> > > ### Author Response · Authors · 2025-11-27
> > >
> > > Thank you for the clarification questions.
> > >
> > > > In what ways does your search for P and for counterexample databases differ from prior approaches, and why are these differences significant?
> > >
> > > The goal of SpotIt is not to heuristically generate databases that can potentially distinguish a predicted SQL from the gold SQL. Instead, it explicitly takes a given predicted SQL and the gold SQL as inputs, and formally proves/disproves their equivalence up to a certain bound. These differences from previous work on synthetic database generation are significant in two ways:
> > > - As an evaluation metric for Text-to-SQL methods, when SpotIt claims the predicted SQL is equivalent to the gold SQL up to size K, it provides a *formal guarantee*. Test-based approaches cannot provide this guarantee.
> > > - SpotIt is guaranteed to find a *minimal* differentiating database for a query pair. Test-based methods cannot provide this guarantee. This minimality is important for easier debugging and is a major benefit of bounded equivalence checking.
> > >
> > > We will clarify this in the related work section.
> > >
> > > > Is the main contribution primarily an extension of VeriEQL to support strings and date operations, or is there a deeper conceptual novelty?
> > >
> > > The main contribution of our paper is threefold. First, we are the first to bring the attention of the Text-to-SQL community to revisiting the test-based evaluation techniques. We show that the current evaluation can be improved in several ways. To achieve this, we leveraged the SOTA method for formal SQL equivalence checking, VeriEQL. Second, we extend SOTA bounded equivalence checking beyond integers and booleans to support common operators that occur in standard Text-to-SQL benchmarks. This required significant engineering effort to extend the verification tool, and theoretical work to extend the soundness proof (Apps. E, F, and G). Third, we present a detailed analysis of the resulting equivalence checks and identify concrete strategies that help developers diagnose and improve gold SQLs.
> > >
> > > We would like to clarify that our primary goal is to develop practical and effective methods to evaluate and analyze Text-to-SQL techniques, which we hope the reviewer agrees is a well-motivated gap to address. We believe our work will be helpful for pushing the Text-to-SQL area forward.
> > >
> > > > The limitations of bounded equivalence remain unclear. Why exactly is the bound k a limiting factor for recursive queries or window functions?
> > >
> > > [**Theoretical limitation of bounded verification**] Bounded equivalence checking has a fundamental limitation in that it can only provide formal guarantees up to the given bound K. The example we gave above is meant to illustrate this point. In theory, bounded checking can establish full unbounded equivalence when the queries satisfy small-scope properties: if a violation exists, it can be witnessed by a counterexample whose size lies below some bound that depends only on the queries. For SQL, determining which fragments satisfy this property and calculating the bound remains an open question.
> > >
> > > In practice, it might be possible to design heuristics to adapt the choice of K based on the terms in the query (e.g., if LIMIT 5 is present, then one should use at least K=6). We are not aware of existing work that proposes a systematic way to do so, but this would be an interesting research direction. At the moment, we believe the best way to determine a suitable bound is through empirical analysis, like in Fig. 4 of the revised version.
> > >
> > > [**Supporting window functions and recursive queries**] We believe it is possible to support these features in bounded equivalence checking. However, it is non-trivial and requires devising novel encodings and soundness proofs, which is a research question in its own right.
> > >
> > > Please let us know if we can provide any additional clarifications.

---

> > > > ### Comment · Reviewer_p5hF · 2025-11-28
> > > > **Thanks**
> > > >
> > > > Thanks for your response. This is an interesting line of work. Extending VeriEQL for bounded equivalence testing of queries in BIRD is not easy, and there are several open questions, some of which are noted in this discussion, that would make valuable additions to the paper. I also hope the authors will make the code available to support reproducibility and future follow-up work.
> > > >
> > > > In light of the authors’ response, I am raising my score.

---

> > > > > ### Author Response · Authors · 2025-11-28
> > > > >
> > > > > Thank you for your thoughtful response and for indicating that you will raise the score. We will incorporate the additional points in our discussion into the revised version. We do plan to release the code to ensure reproducibility and to make SpotIt available as an evaluation metric.

---

### Official Review · Reviewer_eshx · 2025-10-30

**Soundness:** 3
**Presentation:** 4
**Contribution:** 3
**Rating:** 6
**Confidence:** 5

**Summary:**

In the text to SQL systems, evaluation is an important component and the present evaluation methods are largely based on the execution of the generated SQL and ground truth on a static database. Such kind of evaluation may not lead to the best results as two queries can coincidentally produce the same output on the test database while actually being different. SPOTIT proposes a new alternative evaluation pipeline, where a formal bounded equivalence verification engine actively searches for a database that differentiates the generated and ground-truth SQL queries.

**Strengths:**

In text to SQL evaluation plays a vital role and this paper acknowledged the issue. Current methods relying on test-based execution on a static database are optimistic. The proposed solution, SPOTIT, replaces the weak test-based check with a much stronger, formal verification method. This also showed that the reported accuracy of these methods drops by 11.3%–14.2% when switching from the official test-based evaluation to SPOTIT

**Weaknesses:**

The sensitivity of the bound parameter (k) is not analyzed, leaving unclear how verification completeness and runtime scale with k. The paper does not report statistics on timeouts or unsupported queries.

**Questions:**

How is the experiment impacted on changing bound (K) and does increasingly reveal more counter examples.

---

> ### Author Response · Authors · 2025-11-22
> **Thank you for your feedback!**
>
> Please kindly refer to our response to all for the effect of K. Table 2 of the paper reports SpotIt’s runtime and the percentage of supported SQL query pairs.

---

> > ### Author Response · Authors · 2025-11-26
> >
> > Dear reviewer, we hope our responses address your questions and concerns. Please let us know if there is anything else you would like us to clarify. Thank you!
> >
> > Sincerely,
> >
> > Authors

---

### Official Review · Reviewer_NZfv · 2025-11-01

**Soundness:** 4
**Presentation:** 3
**Contribution:** 3
**Rating:** 6
**Confidence:** 3

**Summary:**

Traditional Text-to-SQL evaluation typically relies on comparing execution results. However, this can miss subtle semantic differences between the generated and gold SQL, as different queries may produce the same result on a particular database. SPOTIT addresses this limitation by introducing a formal verification-based evaluation method that uses an SMT solver to construct minimal test databases where two SQLs have different execution results. SpotIt supports richer SQL queries than existing methods. Existing text2SQL methods show a significant performance drop after employing SpotIt, highlighting the need for more rigorous SQL verification.

**Strengths:**

1. This paper tackles an important problem in Text-to-SQL, assessing SQL correctness beyond execution equivalence. By searching for minimal counterexample databases where two SQL queries diverge, SpotIt offers a more rigorous correctness check and greatly simplifies debugging (e.g., identifying faulty ground truths in BIRD).
2. The paper extends the SQL syntax that can be encoded into an SMT solver, which is a great contribution. It consistently covers more than 90 percent of the queries generated by various models.
3. I liked the error analysis the authors did on the BIRD benchmark regarding why the gold SQLs are incorrect. I’ve also personally encountered many erroneous ground truth SQL queries in BIRD, and it used to take me a lot of time to figure out why the reference SQL was wrong. With SpotIt, it would save researchers a significant amount of time identifying these issues.

**Weaknesses:**

1. SPOTIT only searches for divergences up to a fixed tuple-count bound K. Semantic differences that manifest only on larger databases (e.g., join multiplicity) may go undetected if no small counterexample exists.
2. As schema size or query complexity grows (many tables, join many tables together, deep nesting, large numbers of predicates), the SMT solver’s search space can explode, leading to longer solve times.
3. There are still some unsupported SQL features, such as window and analytics functions. I know those might be out of the scope of this paper, but it would be good to discuss them and give examples of SQL queries that can’t be supported.

**Questions:**

1. Did you run any experiments with varying K? How does the latency and accuracy change with increasing K? Would we discover more erroneous SQL queries if we increase K?
2. In practice, how do users choose the tuple-count bound K for counterexample search? Is there a systematic way to trade off between “smallness” and coverage of semantic differences?
3. How does SpotIt latency grow with SQL complexity?
4. On a more complex text2sql benchmark, such as Spider 2.0, could you estimate the percentage of SQL queries that can be covered by SpotIt? It's ok if the percentage is low because not everything can be encoded into SMT solver. But it would be nice to give readers more context because the community is shifting to more complicated benchmarks.

---

> ### Author Response · Authors · 2025-11-22
> **Thank you for your feedback!**
>
> Please kindly refer to our response to all for the effect of K and an evaluation on Spider 2.0 benchmarks.
>
> >  SpotIt may not uncover counterexamples under small bounds.
>
> Incompleteness is indeed one of the fundamental limitations of bounded model checking. Despite this limitation, bounded verification is in practice a widely adopted formal verification technique, as it allows for minimal counterexamples (which is especially helpful in debugging SQL queries) and tends to be more scalable.
>
> We do want to point out that prior work [1] has observed that mistakes in queries can usually be explained by a small number of tuples in practice. This finding is also consistent with our observation that using a larger bound yields diminishing marginal returns. We will clarify this in the revision.
>
> [1] Miao et al. Explaining wrong queries using small examples. SIGMOD’19.
>
> > How to choose a bound for verification?
>
> Choosing the optimal bound for bounded verification is in general very challenging. It might be possible to design heuristics to adapt the choice of K based on the terms in the query (e.g., if LIMIT 5 is present, then one should use at least K=6). We are not aware of existing work that proposes a systematic way to do so, but this would be an interesting research direction. At the moment, we suggest to determine a suitable bound with empirical analysis, like in Fig. 4 of the revised version.
>
> > How does SpotIt latency grow with SQL complexity?
>
> We performed such an analysis and report how the runtime changes with respect to the number of columns, integrity constraints, tables, sub-queries in the gold SQL, and nodes in the abstract syntax tree in the gold SQL. We found that all parameters, except for the number of tables, are positively correlated with the median runtime.
>
> | # columns | Median runtime (s) |
> |-------------|---------------------|
> | 11          | 0.4310              |
> | 21          | 0.1701              |
> | 31          | 0.1963              |
> | 48          | 0.3887              |
> | 55          | 0.5576              |
> | 64          | 0.3007              |
> | 71          | 0.5365              |
> | 89          | 0.2672              |
> | 94          | 0.3154              |
> | 115         | 0.6537              |
> | 199         | 0.8006              |
>
> | # constraints | Median runtime (s) |
> |------------------|---------------------|
> | 5                | 0.2842              |
> | 7                | 0.1701              |
> | 10               | 0.4324              |
> | 16               | 0.5576              |
> | 17               | 0.3887              |
> | 19               | 0.1963              |
> | 21               | 0.5365              |
> | 36               | 0.6877              |
>
> | # tables | Median runtime (s) |
> |-------------|---------------------|
> | 3           | 0.2842              |
> | 4           | 0.4310              |
> | 5           | 0.1701              |
> | 6           | 0.6537              |
> | 7           | 0.8006              |
> | 8           | 0.4249              |
> | 10          | 0.1963              |
> | 13          | 0.3154              |
>
>
>
>
> | # subqueries | Median runtime (s) |
> |----------------|---------------------|
> | 0              | 0.3676              |
> | 1              | 0.3972              |
> | 2              | 1.9128              |
>
>
> | # AST nodes | Median runtime (s) |
> |------------|---------------------|
> | 0–19       | 0.1812              |
> | 20–39      | 0.3408              |
> | 40–59      | 0.3426              |
> | 60–79      | 0.4216              |
> | 80–99      | 0.9683              |
> | 100–119    | 1.1865              |
> | 120–139    | 0.2053              |
> | 140–159    | 1.2395              |

---

> > ### Author Response · Authors · 2025-11-26
> >
> > Dear reviewer, we hope our responses address your questions and concerns. Please let us know if there is anything else you would like us to clarify. Thank you!
> >
> > Sincerely,
> >
> > Authors

---

### Official Review · Reviewer_E2Yh · 2025-11-04

**Soundness:** 3
**Presentation:** 2
**Contribution:** 3
**Rating:** 4
**Confidence:** 4

**Summary:**

The paper introduces SpotIt, a verification-based evaluation pipeline for Text-to-SQL that replaces single-database execution tests with SMT-based bounded equivalence checking. Building on VeriEQL, the authors extend the supported SQL fragment with dates/strings, implicit type casts, set-semantics equivalence, and provide correctness proofs for new encodings. On BIRD-dev (1,533 items) across 10 top-performing systems, accuracy reported by standard EX-TEST drops by roughly 9.8–13.5% under SpotIt/SpotIt+, and rankings change substantially. Minimal counterexample databases expose three root causes for mismatches: incorrect gold SQL, ambiguous NL questions, and incorrect generated SQL. Manual audits suggest problematic gold SQLs are common; in some cases, ambiguity makes multiple SQLs “reasonable”. Algorithms for iterative bounds, validation against a real DBMS to filter spurious CEX, and cross-checking of counterexamples across systems are described. Runtime is practical (seconds per pair) with good coverage gains over baseline VeriEQL.

**Strengths:**

S1. Strong and timely problem framing; evaluation reliability is critical for Text-to-SQL progress.
S2. Clean pipeline with validation and cross-checking; minimal CEXs make root causes transparent.
S3. Substantive empirical findings (accuracy/ranking shifts; gold SQL issues; ambiguity prevalence).
S4. Formal treatment and proofs for extended string/date operators and set-semantics equivalence.
S5. Practicality: seconds-scale per-instance runtime; high coverage relative to prior verifier.

**Weaknesses:**

W1. Bound sensitivity (K): The paper does not quantify how many non-equivalences require bounds beyond K=5, so the false-negative rate is unknown. A detection-vs-K curve and examples missed at K=5 are needed to calibrate trust in results.
W2. Dataset scope: All results are on BIRD-dev, leaving external validity uncertain. Including Spider 2.0 and one enterprise-style schema would test generality and robustness of conclusions.
W3. Cause attribution (gold vs ambiguity vs model): The labeling relies on a small, non-adjudicated manual audit, which risks subjectivity. A larger, blinded, multi-annotator study with agreement stats would strengthen these claims.
W4. DBMS semantics alignment: The SMT encodings may not match engine-specific behaviors (NULLs, casts, date ops), but this isn’t systematically evaluated. Cross-engine validation (SQLite/MySQL/Postgres) and documenting divergences are necessary.
W5. Presentation quality: Frequent typos/formatting glitches and occasional terminology inconsistencies reduce clarity (e.g., “SQL ITE”, “hosptial”). A thorough proofreading and added navigational aids for dense formal sections would improve readability.

**Questions:**

1. How sensitive are SpotIt’s findings to the choice of K? Any empirical curve of detected non-equivalences vs K and the diminishing returns?
2. Did you observe cases where K=5 missed mismatches that appeared at higher K? Any characteristic patterns?
3. How many spurious CEXs did the validation filter out, and which operators were most responsible?
4. Can you report results on Spider 2.0 (or BIRD-Interact) to assess generalization beyond BIRD-dev?
5. How do encodings reconcile engine idiosyncrasies (e.g., string-to-int casting behavior, date boundaries)? Any cross-engine discrepancies observed?
6. For ambiguity cases, would adding light-weight interaction (à la BIRD-Interact) change pass/fail judgments under SpotIt?

---

> ### Author Response · Authors · 2025-11-22
> **Thank you for your feedback!**
>
> Please kindly refer to our response to all for the effect of K and an evaluation on Spider 2.0 benchmarks.
>
> > How many spurious CEXs did the validation filter out, and which operators were most responsible?
>
> We report the percentage of counterexamples that were successfully validated for each method. Overall, 93.34%-96.15% of the counter-examples are non-spurious, which suggests our encoding is usually sufficiently precise. We have extended Table 2 of the paper with these results.
>
> | Method (# quest.) | Valid. (%) |
> |-------------------|------------|
> | Alpha       | 96.15      |
> | Chess  | 93.34      |
> | CSC-32b | 94.46      |
> | CSC-7b | 95.10    |
> | GENA-1 | 94.52     |
> | GENA-2 | 95.42     |
> | GSR    | 94.86      |
> | Omni-Maj | 95.83      |
> | RSL  | 95.62      |
> | SLM | 95.05      |
>
>
>
> The main source of spurious counterexamples arises from the non-deterministic semantics of the ORDER BY operator when its ordering expression does not fully determine a unique ordering of the rows. In particular, if two or more rows share identical values for all expressions in the ORDER BY clause, the query is permitted to return those rows in any relative order.
>
> For example, consider the table T:
>
> | name | age |
> |------|-----|
> | A    | 30  |
> | B    | 30  |
>
> The query SELECT * FROM T ORDER BY age LIMIT 1 may return either (A, 30) or (B, 30) as the result.
>
> However, in reality, database systems apply their own internal heuristics to break such ties, which is prohibitively difficult to predict. This means that a counterexample deemed valid by VeriEQL based on some tie-breaking rule may be actually spurious.
>
> > How to handle discrepancies across different engines and SQL dialects? How do encodings reconcile engine idiosyncrasies (e.g., string-to-int casting behavior, date boundaries)? Any cross-engine discrepancies observed?
>
> We have observed several discrepancies across different database engines. For example, MySQL supports dates in the range '1000-01-01' to '9999-12-31' [1], whereas SQLite supports dates from '0000-01-01' to '9999-12-31' [2].  Other differences arise in how various SQL dialects handle date modifiers and nested queries. Fortunately, each dialect provides a comprehensive list of idiosyncrasies in their documentation. SpotIt is parameterized by the SQL dialect and therefore generates SMT encodings that are tailored to the specific dialect. In our evaluation, we found this is important for improving the precision of our verification results.
>
> [1] https://dev.mysql.com/doc/refman/9.4/en/datetime.html
>
> [2] https://sqlite.org/lang_datefunc.html
>
>
> > For ambiguous cases, would adding light-weight interaction (à la BIRD-Interact) change pass/fail judgments under SpotIt?
>
> We believe that extending the text-to-SQL task beyond a one-shot prediction setting is a sensible way to deal with ambiguity. However, Spotit is orthogonal to such an extension, as any evaluation would eventually involve checking whether a prediction matches the gold SQL. SpotIt is proposed as an alternative way to perform this check.
>
>
> > The labeling relies on a small, non-adjudicated manual audit, which risks subjectivity.
>
> Each label was examined by and agreed upon by at least two annotators, which aligns with the community standard.

---

> > ### Author Response · Authors · 2025-11-26
> >
> > Dear reviewer, we hope our responses address your questions and concerns. Please let us know if there is anything else you would like us to clarify. Thank you!
> >
> > Sincerely,
> >
> > Authors

---

### Author Response · Authors · 2025-11-12
**Paper revision plan**

Dear reviewers,

Thank you for the constructive feedback! We are happy that the reviewers find our approach towards Text-to-SQL evaluation timely and important, and our findings and analysis interesting and practically meaningful.

There are two main questions that the reviewers ask for clarification:
- What is the effect of the choice of the bound K?
- How does SpotIt perform on additional benchmarks, such as Spider 2.0?

We agree with the reviewers that answering these questions would further strengthen our evaluation. We will run experiments and report results within the rebuttal window.

We will also address reviewers' individual comments and present additional results when appropriate.

Sincerely,

Authors

---

### Author Response · Authors · 2025-11-22
**Response to all: effect of K, results on Spider 2.0, clarification of (un)supported features**

Dear reviewers,

As promised earlier, we report here
- the effect of the verification bound K, and
- experimental results on Spider 2.0 benchmarks.

In addition, we clarify the SQL features that SpotIt can and cannot support.

We provide individual responses to reviewers' other comments and questions.

## Effect of the verification bound K

To study the effect of the verification bound K, we varied its value from 1 to 7 and run SpotIt on the predictions of CSC-32b, the best model on BIRD according to the EX-test. As shown in the table below, SpotIt was able to find substantially more differentiating databases when K increases from 1 to 2, and there are only marginal gains past K=3. This justifies our choice of K = 5. The additional inconsistencies using K=6 were all discovered within 10 seconds.

| Method            | accuracy |
|------------------|----------|
| EX-test          | 71.32    |
| SpotIt (K=1)     | 64.41    |
| SpotIt (K=2)     | 59.45    |
| SpotIt (K=3)     | 59.00    |
| SpotIt (K=4)     | 58.87    |
| SpotIt (K=5)     | 58.80    |
| SpotIt (K=6)     | 58.60    |
| SpotIt (K=7)     | 58.60    |

We have incorporated this result and a detection-vs.-K curve as suggested by Reviewer E2Yh in Sec. 5.

**Examples of inconsistency that can only be detected with larger K.**
Examples D.3 and D.8 in the papers show two query pairs that remain equivalent for K=1, but are non-equivalent for K=2. In D.3, this is due to the fact that one SQL query looks for the maximal element (ASC) while the other searches for the minimal element (DESC), which would obtain the same result if there is only one row in the table. In D.8 this is due to the fact that one query counts the number of DISTINCT elements, while the other allows repetition. As an additional example, SpotIt found the following pair of queries equivalent for K$\leq$5 but non-equivalent for K=6:
- *Gold*: SELECT T1.full_name FROM superhero AS T1 INNER JOIN race AS T2 ON T1.race_id = T2.id WHERE T2.race = 'Demi-God'
- *Prediction*: SELECT s.full_name FROM superhero s JOIN race r ON s.race_id = r.id WHERE r.race = 'Demi-God' LIMIT 5


## Evaluation on Spider 2.0 benchmarks.

To further examine the generalizability of SpotIt, we evaluate it on Spider 2.0 benchmarks. We use a state-of-the-art Text-to-SQL method OmniSQL and GPT-5 to generate queries for the 135 SQLite benchmarks. They passed EX-Test on 34.1% and 42.2% questions, respectively, which is competitive with top entries on the Spider 2.0 leaderboard. As shown in the table below, SpotIt finds 16 and 8 query pairs that are deemed correct by testing-based evaluation respectively for OmniSQL and GPT-5. This is consistent with our finding that SpotIt can uncover query discrepancies overlooked by test-based evaluation. Overall, SpotIt can efficiently identify counterexamples for more complex queries from Spider 2.0.

| Model       | Acc. (EX-test) | Acc. (SpotIt) | Supported (%) | Avg. Time (s) |
|-------------|--------------|---------------------|----------------|-----------|
| OmniSQL     | 34.1%        | 22.2%               | 60.9%          | 3.4  |
| GPT-5       | 42.2%         | 36.3%                | 50.9%          | 1.1  |

SpotIt’s runtime for finding counterexamples remains low. We believe this is due to the fact that the schemas in Spider 2.0 are only moderately larger than those in BIRD (78.6 -> 97.6 columns) and counter-examples can usually be detected with small values of K. However, we did find that there are more Spider 2.0 queries that include operators currently unsupported by SpotIt. In particular, 52.6% of unsupported queries involve the window function (i.e., the OVER clause).  We have added this analysis, along with additional discussion, in Section 5 of the revised version.


## Clarifying the SQL features that SpotIt can or cannot support.

To our best knowledge, among existing SQL equivalence verifiers, VeriQEL supports the most expressive subset of SQL, including nested subqueries, HAVING clauses, and complex joins. In this paper, we extended it to support a variety of string and date operations. To tackle queries from Spider 2.0, we further support the Contain and Concat functions for strings, and the DataShift function for dates. We have updated Fig. 2 of the paper to reflect this extended SQL syntax, as well as the soundness proof in Apps. E, F, and G.

Our verification engine currently does not support window functions or recursive common table expressions, which may appear in some Text-to-SQL benchmarks. In addition, we currently do not support operations over semi-structured data (e.g., arrays and geometries), for which equivalence checking is inherently challenging.

In the revised version, we added a detailed summary about the limitations of our verification engine in App. C and referred to it in the main paper.

---

### Meta-Review · Area_Chair_7Ru2 · 2026-01-08

**Summary:**

This paper presents a timely framework, SPOTIT, a verification-based evaluation pipeline for Text-to-SQL that replaces single-database execution tests with SMT-based bounded equivalence checking.

Across reviewers, a primary concern is bound sensitivity and completeness. Multiple reviewers note that SPOTIT’s reliance on a fixed tuple bound K (e.g., K=5) is insufficiently analyzed. The paper does not quantify how many semantic divergences require larger bounds, nor does it report false-negative rates, timeout statistics, or detection–vs.–K trade-offs. As a result, it remains unclear how verification completeness and runtime scale with increasing query or schema complexity.

Another major issue is external validity and dataset scope. All experiments are conducted solely on BIRD-dev, leaving open questions regarding generalization to other benchmarks (e.g., Spider 2.0) or more complex, enterprise-style schemas. Reviewers suggest that broader evaluation across datasets and database engines would be necessary to substantiate the paper’s general claims.

Reviewers also raise concerns about cause attribution and annotation rigor. The current labeling of errors (e.g., gold error vs. ambiguity vs. model error) relies on a small, non-adjudicated manual audit, which introduces subjectivity. A larger, blinded, multi-annotator study with inter-annotator agreement metrics would significantly strengthen these conclusions.

In the rebuttal, the authors added additional experiments to systematically analyze the effect of the verification bound K and evaluated the proposed method on broader datasets to assess its generalizability. I lean toward a weak accept, as the authors have addressed most of the concerns raised by the reviewers and clarified what the proposed method can and cannot do. In doing so, they reveal previously underexplored limitations of existing state-of-the-art evaluation platforms and demonstrate that their approach can serve as an effective complementary method to improve current evaluation frameworks.

**Reviewer Concerns:**

I think one of the primary concerns has been partially addressed in the rebuttal. The authors provide additional experiments analyzing the effect of the verification bound K and further examine the generalizability of SPOTIT by evaluating it on the Spider 2.0 benchmark. These additions help clarify the impact of the bound choice and strengthen the evidence for cross-dataset applicability.

However, reviewers’ concerns regarding scalability and deeper failure-case analysis remain unaddressed. In particular, the rebuttal does not sufficiently analyze solver runtime behavior as query and schema complexity increase, nor does it provide a systematic characterization of hard or missed cases. As such, these issues continue to limit confidence in the framework’s robustness and practical applicability.

**Reviewer Scores:**

Reviewer E2Yh: I believe the rebuttal does not substantially change this reviewer’s position, and therefore they would likely maintain their original score of 4.

Reviewer NZfv: Although no explicit response was provided, I think the additional experiments sufficiently address this reviewer’s primary concerns. As such, I expect the score to remain at 6.

Reviewer eshx: This reviewer did not provide a follow-up response. In my view, the authors’ additional analysis on the verification bound K, along with the reported runtime statistics for unsupported queries, addresses the main issues raised. I therefore expect the score to remain at 6.

Reviewer p5hF: This reviewer indicated that the rebuttal satisfactorily addressed their concerns and increased their score from 4 to 6.

---

### Decision · Program_Chairs · 2026-01-26

Accept (Poster)